# Strong and Weak Identifiability of Optimization-based Causal Discovery in Non-linear Additive Noise Models

**Mingjia Li** [1]  **Hong Qian*** [1]  **Tian-Zuo Wang** [2][3]  **Shujun Li** [1]  **Min Zhang** [1]  **Aimin Zhou** [1]

## Abstract

Causal discovery aims to identify causal relationships from observational data. Recently, optimization-based causal discovery methods have attracted extensive attention in the literature due to their efficiency in handling high-dimensional problems. However, we observe that optimization-based methods often perform well on certain problems but struggle with others. This paper identifies a specific characteristic of causal structural equations that determines the difficulty of identification in causal discovery and, in turn, the performance of optimization-based methods. We conduct an in-depth study of the additive noise model (ANM) and propose to further divide identifiable problems into strongly and weakly identifiable types based on the difficulty of identification. We also provide a sufficient condition to distinguish the two categories. Inspired by these findings, this paper further proposes GENE, a generic method for addressing strongly and weakly identifiable problems in a unified way under the ANM assumption. GENE adopts an order-based search framework that incorporates conditional independence tests into order fitness evaluation, ensuring effectiveness on weakly identifiable problems. In addition, GENE restricts the dimensionality of the effect variables to ensure *scale invariance*, a property crucial for practical applications. Experiments demonstrate that GENE is uniquely effective in addressing weakly identifiable problems while also remaining competitive with state-of-the-art causal discovery algorithms for strongly identifiable problems.

[1]Shanghai Institute of AI Education, and School of Computer Science and Technology, East China Normal University, Shanghai, China. [2]National Key Laboratory for Novel Software Technology, Nanjing University, Nanjing, China. [3]School of Artificial Intelligence, Nanjing University, Nanjing, China. Correspondence to: Hong Qian <hqian@cs.ecnu.edu.cn>.

*Proceedings of the $42^{nd}$ International Conference on Machine Learning*, Vancouver, Canada. PMLR 267, 2025. Copyright 2025 by the author(s).

## 1. Introduction

Discovering causal relationships behind variables is crucial to data science and plays a key role in a wide range of fields, including telecommunication network fault diagnosis (Li et al., 2024), education (Schochet, 2013) and human ethology (Cai et al., 2017). While randomized controlled trials (RCTs) are considered the gold standard for studying causal relationships, they are usually expensive or even impossible to implement. This limitation motivates causal discovery methods that can infer causal relationships from observational data. Recent advances in this field have made causal discovery an increasingly vital topic (Spirtes et al., 2000; Pearl, 2009).

Optimization-based causal discovery is a class of methods that aims to identify causal relationships from observational data by leveraging optimization techniques. The core idea behind optimization-based causal discovery is to first assign a fitness score to a given causal structure based on how well it fits the observed data. Optimization techniques are then employed to search for causal graphs with high fitness scores, thereby facilitating the discovery of causal relationships. In recent years, continuous-optimization-based causal discovery methods (Zheng et al., 2018; 2020; Lachapelle et al., 2020; Yu et al., 2019; Liu et al., 2024) have garnered extensive attention in the literature due to their capability to deal with high-dimensional problems efficiently, making them particularly appealing for large-scale data analysis.

However, we observe that even under the same causal model assumption, namely, the additive noise model (ANM) (Hoyer et al., 2008), optimization-based causal discovery may perform very differently. Figure 1 presents two pairs of such examples (a) and (b), where in each pair, the causal models above and below are both under ANM and share the same causal structures. For Figure 1 (a.1), where $y = x^2 + N$ and $N$ is a noise term, it is easy to identify the true causal structure using only regression, as regression along the correct causal direction fits well but not vice versa. In contrast, for Figure 1 (a.2), where $y = x^3 + x + N$, regression generally fits well for both directions. Distinguishing the fitting difference between the two directions may require an extremely large amount of data; otherwise, a wrong causal graph is likely to be learned with limited samples. In

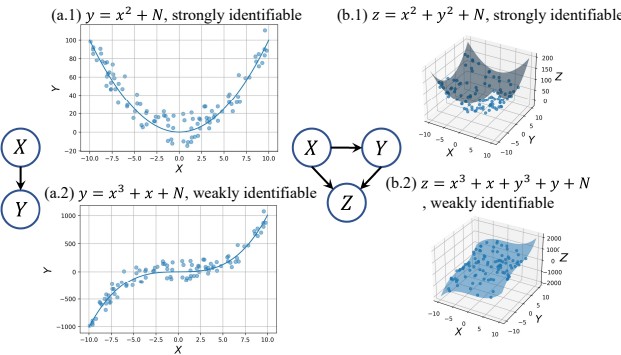

*Figure 1.* Examples of the strongly and the weakly identifiable causal models. Here $N$ is a Gaussian noise term.

such cases, additional properties must be utilized (Hoyer et al., 2008; Zhang & Hyvärinen, 2009). Similar situations apply to the examples in Figure 1 (b.1) and (b.2). These examples demonstrate that even for identifiable causal models with the same causal structure, the difficulty of identification can vary significantly.

We highlight that this difference arises from the existence of implicit functions within the structural equations of the causal model. These implicit functions categorize problems into strongly and weakly identifiable types. Additionally, we derive a sufficient condition to distinguish between these categories. Specifically, strongly identifiable models are easier to identify, as they yield a clear distinction in the regression process metrics between different causal directions. Conversely, weakly identifiable models lack this distinction due to the existence of implicit functions, making them more difficult to identify. With limited samples, additional criteria, such as residual independence (Hoyer et al., 2008; Zhang & Hyvärinen, 2009), are necessary in the identification process of weakly identifiable models.

Unfortunately, the strength of identifiability can rarely be known in advance. To address both strongly identifiable and weakly identifiable problems in a unified manner, this paper proposes GENE: a generic causal discovery method for both strongly and weakly identifiable problems under ANM. GENE employs an order-based search framework that integrates conditional independence tests into the fitness evaluation of the order, which is vital for enhancing its effectiveness in addressing weakly identifiable problems. Interestingly, our results align well with (Reisach et al., 2021), which points out that certain continuous optimization methods may suffer from data rescaling processes, such as data standardization and changing the units of variables; furthermore, we reveal that this limitation of continuous optimization methods arises because their optimization process does not explicitly distinguish between cause and effect in causal relationships, making it unable to control the data scale during fitting. In contrast, GENE restricts the dimen-

sionality of the effect variable by exploiting $R^2$ instead of mean square error (MSE) during order estimation. By doing so, it ensures scale invariance [1], meaning it performs stably under different data scales, which is a crucial attribute in practical applications. By leveraging these advantages, GENE provides a robust solution for causal discovery across a wide range of scenarios. Refer to Appendix A for a detailed review of related work.

Extensive experiments are conducted on both synthetic and real-world Sachs (Sachs et al., 2005) datasets. The results indicate that, compared to the state-of-the-art algorithms (SOTAs), only GENE is effective for weakly identifiable problems, while for the strong ones, GENE remains competitive. It is worth noting that, on the real-world Sachs dataset, GENE is significantly superior to SOTAs.

## 2. Preliminaries

Before delving into the specifics of our approach, we first outline the foundational concepts for understanding causal discovery, identifiability, and order-based methods.

### 2.1. Causal Discovery

Causal discovery, also known as causal structure learning, aims to discover the graph of the causal graphical model (CGM) $M = (P_v, \mathcal{G})$ behind the observational data. Our work primarily considers scenarios in the absence of latent variables (though see (Wang et al., 2023; Ni et al., 2025) for approaches addressing causal discovery with latent variables). Formally, given the sample matrix $D \in \mathbb{R}^{n \times d}$ with $n$ i.i.d. observations sampled from the joint probability distribution $P_v$ over a variable set of $d$ variables $V = \{V_1, V_2, \ldots, V_d\}$, the task is to find a directed acyclic graph (DAG) $\mathcal{G} = (V, E)$ which best describes $P_v$. A structural equation model (SEM) is defined as $\mathcal{S} = (S, P_n)$. $S$ is a set of $d$ structural equations, and $P_n$ is the joint probability distribution of mutually independent noise variables $N = \{N_1, N_2, \ldots, N_d\}$, i.e.,

$$S_i : V_i = f_i(V_{\text{pa}(i)}, N_i), i = 1, 2, \ldots, d, \quad (1)$$

where $V_{\text{pa}(i)}$ denotes the parents of node $i$ in $\mathcal{G}$, and $f_i$ is a function $\mathbb{R}^{|V_{\text{pa}(i)}|+1} \to \mathbb{R}$. Like CGM, SEM also entails $P_v$, as sampling from $P_v$ is equivalent to sampling from $P_n$ and then propagating the samples through $S$.

### 2.2. Causal Identifiability

**Definition 2.1** (Identifiability). Given a set of assumptions $A$ on a CGM $M = (P_v, \mathcal{G})$, graph $\mathcal{G}$ is said to be identifiable from $P_v$ if there exists no other CGM $\hat{M} = (\hat{P}_v, \hat{\mathcal{G}})$ satisfying all assumptions in $A$ such that $\hat{\mathcal{G}} \neq \mathcal{G}$ and $\hat{P}_v = P_v$.

---

[1]Details are discussed in the Causal Order Estimate section.

Generally speaking, the identifiability of a CGM is determined by whether multiple distinct DAGs can represent the same probability distribution $P_v$; if so, these DAGs are said to belong to the same equivalence class. In fact, under causal faithfulness and Markov assumptions, causal discovery often only recovers a graph up to its Markov equivalence class (MEC) (Chickering, 1995; Spirtes et al., 2000; Wang et al., 2024). There are many studies on the causal identifiability under some specific scenarios, such as linear functions with non-Gaussian noise (LiNGAM) (Shimizu et al., 2006), non-linear functions with additive noise (ANM) (Hoyer et al., 2008), and post-nonlinear functions (Zhang & Hyvärinen, 2009). This paper follows the ANM setting with the SEM written as:

$$V_i = f_i(V_{\text{pa}(i)}) + N_i, i = 1, 2, \ldots, d, \tag{2}$$

where $f_i$ is a non-linear function from $\mathbb{R}^{|V_{\text{pa}(i)}|} \to \mathbb{R}$. In ANM, (Hoyer et al., 2008) show that for bivariate case, i.e., only consider two variables $X$ and $Y$, identifiability is accessible through residual independence analysis. Specifically, we can first apply a non-linear regression of $Y$ on $X$, and calculate the residual between the predicted value and the true value, then test the independence of the residual and the cause variable (here is $X$). Do the same thing in another direction (regress $X$ on $Y$). If the regression direction is correct, the residual should be independent of the cause variable, and this does not hold if the direction is incorrect under some mild assumptions (Hoyer et al., 2008). Thus the causal direction between $X$ and $Y$ is identifiable.

### 2.3. Order-Search-based Causal Discovery

It is well known that for a CGM, there exists a causal order $\pi$ defined on the variable set $V = \{V_1, V_2, ..., V_d\}$. This order is a permutation of magnitude $d$ that specifies the cause-effect relation: Only a preceding variable in the order can be the parent (cause) of a subsequent variable and not conversely. It is worth noting that there is an intrinsic connection between orders and complete DAGs (also known as fully connected DAGs): for any order $\pi$, we can construct a complete DAG $C^\pi$, where each variable $V_{\pi(i)}$ has a directed edge connecting to all $V_{\pi(j)}$ with $i < j, i, j = 1, 2, \ldots, d$. Besides, for a given DAG $\mathcal{G}$, we define the true order set as:

$$\Pi^{\mathcal{G}} = \{\pi \mid \text{the complete DAG } C^\pi \text{ is a super-DAG of } \mathcal{G}\},$$

where the term a super-DAG of $\mathcal{G}$ denotes a DAG that the edge set of $\mathcal{G}$ is a subset of the edge set of this DAG. Evidently, the true order for any incomplete $\mathcal{G}$ is not necessarily unique, i.e., the magnitude of $\Pi^{\mathcal{G}}$ is possibly greater than 1.

Order-based causal discovery is motivated by the following fact. If the causal order among the variables is known, the remaining task is simply variable selection, which can be handled by multivariate regression. Therefore, the problem reduces to estimating and searching for the causal order. This idea can at least date back to Teyssier & Koller (2005); Schmidt et al. (2007). First, compared to DAG space search, order-based methods deal with a much smaller search space. Second, it avoids the issue of enforcing acyclicity since the order naturally guarantees no cycles in the graph. Third, the causal order is closely related to identifiability since they both focus on the cause-effect direction, which means that the existing identifiability techniques can be incorporated into order-based methods.

## 3. Strength of Identifiability

In this section, we propose a formal definition for the strength of identifiability, dividing SEMs into strongly and weakly identifiable ones. We further disclose a sufficient condition to distinguish between strong and weak identifiability, which is proved based on the implicit function theorem (Rudin et al., 1976).

Given an SEM with its structural equation set $S = \{S_1, S_2, \ldots, S_d\}$ that follows the assumptions of ANM as in Eq. (2), we consider the structural equation $S_i$, which describes the causal relation between the parent set $V_{\text{pa}(i)}$ and $V_i$. It can be rewritten as:

$$
\begin{aligned}
& V_i = f_i(V_{\text{pa}(i)}) + N_i \,, \\
\Rightarrow \ & V_i - f_i(V_{\text{pa}(i)}) - N_i = 0 \,, \\
\Rightarrow \ & F_i(V_i, \text{pa}(i)_1, \text{pa}(i)_2, \ldots, \text{pa}(i)_j, N_i) = 0 \,, \quad (3)
\end{aligned}
$$

where $\text{pa}(i)_j$ denotes the $j$th parent of $V_i$. Eq. (3) is defined as the equivalent implicit equation of $S_i$. Now we introduce the definition of the strength of identifiability.

**Definition 3.1** (Strength of Identifiability). Given an SEM $\mathcal{S}$ following the ANM assumptions, with its structural equation set $S = \{S_1, \ldots, S_d\}$, we consider $S_i$ and its equivalent implicit equation as in Eq. (3). If for every $\text{pa}(i)_k \in V_{\text{pa}(i)}$, $F_i$ cannot give an implicit function as:

$$\text{pa}(i)_k = g(V_i, \text{pa}(i)_1, \ldots, \text{pa}(i)_{k-1}, \text{pa}(i)_{k+1}, \ldots, \text{pa}(i)_j, N_i) \,,$$

where $g(\cdot)$ denotes a function from $\mathbb{R}^{|V_{\text{pa}(i)}+1|} \to \mathbb{R}$, that is, if such an implicit function does not exist for every $\text{pa}(i)_k$, then $S_i$ is defined as *strongly identifiable*. Otherwise, it is *weakly identifiable*. Moreover, if all $S_i \in S$ are strongly identifiable, $\mathcal{S}$ is *strongly identifiable*; otherwise, $\mathcal{S}$ is *weakly identifiable*.

According to Definition 3.1, the strength of identifiability is determined by the existence of an implicit function. If an implicit function for the parent variables does not exist, the SEM can be identified through simple regression. Otherwise, we need to check the independence between the regression residual and the regressors. However, in practice, it is still challenging to distinguish between strong and weak

identifiability because we often do not have access to the analytic expression of $F$. Therefore, we further provide a sufficient condition for distinguishing between strong and weak identifiability, which is related to the implicit function theorem (Rudin et al., 1976). Let $x$ denote $(x_1, x_2, \ldots, x_n)$.

**Lemma 3.2** (Global Existence of Multivariate Implicit Function). *Let* $F(x_1, x_2, \ldots, x_n, y)$ *be continuous on the set* $E = \{(x_1, x_2, \ldots, x_n, y) \in \mathbb{R}^{n+1} \mid a \leq x_1, x_2, \ldots, x_n \leq b, -\infty < y < \infty\}$. *Suppose that the partial derivative function* $F_y(x_1, x_2, \ldots, x_n, y)$ *exists for all* $(x_1, x_2, \ldots, x_n, y) \in E$, *and there exist positive constants* $0 < m < M$ *s.t.*

$$m \leq F_y(x_1, x_2, \ldots, x_n, y) \leq M, \ \forall (x_1, x_2, \ldots, x_n, y) \in E,$$

*then the equation* $F(x_1, x_2, \ldots, x_n, y) = 0$ *has a solution* $y = \varphi(x_1, x_2, \ldots, x_n)$ *which is continuous over* $[a, b]^n$.

**Proof Sketch:** To prove the existence and uniqueness of a continuous solution $y = \varphi(x)$ to the equation $F(x, y) = 0$, we define an operator $T$ on the space of continuous functions $C[a, b]^n$. By applying the Lagrange mean value theorem (Rudin et al., 1976) and the assumptions on $F$, we show that $T$ is a contraction mapping. The Banach fixed point theorem (Rudin et al., 1976) then guarantees the existence of a unique fixed point of $T$, which corresponds to the solution $\varphi(x)$. The complete proof is in Appendix B.

With Lemma 3.2, we derive a sufficient condition to determine the strength of identifiability, as Theorem 3.3 below.

**Theorem 3.3** (Sufficient Condition for the Strength of Identifiability). *Given a structural equation* $S_i$ *and its equivalent implicit equation* $F_i(V_i, \mathrm{pa}(i)_1, \ldots, \mathrm{pa}(i)_j, N_i) = 0$, *if there is a* $\mathrm{pa}(i)_k$ *such that* $F_i$ *and* $\mathrm{pa}(i)_k$ *satisfy the condition for* $F$ *and* $y$ *in Lemma 3.2,* $S_i$ *is weakly identifiable.*

Theorem 3.3 can be proved using Definition 3.1 and Lemma 3.2. It indicates that a structural equation is weakly identifiable under a mild condition on the partial derivatives of its equivalent implicit equation. Theorem 3.3 serves as an operational condition to determine the strength of identifiability of a given structural equation. We refer to it as "operationa" because this theorem only requires knowledge of the partial derivatives of the equation without needing its analytic expression. It is worth noting that this condition is not necessary because the implicit function may exist even if the condition does not hold. For example, consider the equation $F(x, y) = y^3 - x$. At the point $(x, y) = (0, 0)$, the partial derivative $F_y = 0$. Consequently, the constant $m$ does not exist. However, the equation $F(x, y) = 0$ still has a solution, which is obviously $y = \varphi(x) = \sqrt[3]{x}$.

# 4. A Generic Approach to Strong and Weak Identifiability in ANM

In this section, we introduce GENE, a method designed to address both strongly and weakly identifiable problems in a unified manner. Broadly, GENE operates in two phases: order search and parent search. The order search phase is further divided into order estimation and optimization. Next we give details of these steps and discuss how GENE effectively overcomes the limitations of existing methods.

## 4.1. Causal Order Estimate

In this subsection we present the first step of GENE: estimating causal orders. First, for any true order of $\mathcal{G}$, there exists a upper triangular representation of Eq. (2) (Bühlmann et al., 2014). Formally, given an order $\pi$, the SEM equation of Eq. (2) can be rewritten as:

$$V_{\pi(i)} = f_i(V_{<\pi(i)}) + N_i, i = 1, 2, \ldots, d, \qquad (4)$$

where $V_{<\pi(i)} = \{V_j : \pi(j) < \pi(i)\}$ refers to the variables preceding $V_i$ in the causal order $\pi$. To estimate such an order, the first aspect that we should consider is how well the preceding nodes in the order can be used to predict subsequent nodes, i.e., the goodness of fit. This task can be handled using log-likelihood (consider $f_i$s and $N_i$s as parameters, then apply maximum likelihood estimate), as in CAM (Bühlmann et al., 2014). In this paper, we propose to apply $R^2$, a metric of goodness of fit ranging from 0 to 1, to evaluate the quality of regression. More specifically, for each node $V_{\pi(i)}$, we train a fully connected neural networks (NN) with 2 hidden layers of size $h$, parameterized by $\sigma(i) = \{W^1, W^2\}$ where $W^1$ and $W^2$ are the weight matrix. The NN takes $\{V_{\pi(1)}, ..V_{\pi(i-1)}\} \in \mathbb{R}^{i-1}$ as input to predict $V_{\pi(i)}$. In other words, we use a NN to fit the potential function between $V_{<\pi(i)}$ and $V_i$, and the degree of fitness is evaluated by

$$R^2(V_{\pi(i)}) = 1 - \frac{\mathrm{MSE}(\hat{f}_i(V_{<\pi(i)}), V_{\pi(i)})}{\mathrm{Var}(V_{\pi(i)})}, \qquad (5)$$

where $\hat{f}$ denotes the approximated function, $\mathrm{Var}(V_{\pi(i)})$ is the variance of $V_{\pi(i)}$ and MSE refers to the mean square error between the predictions and the true values of $V_{\pi(i)}$.

It is worth noting that continuous-optimization-based methods (details in Appendix A.2) perform differently after data rescaling (Reisach et al., 2021; 2023), whereas GENE is scale invariant. This is because these methods encode the entire SEM into a connectivity matrix and then minimize the overall MSE, implying that they are not able to explicitly distinguish between cause and effect during optimization. Consequently, rescaling affects the MSE. In contrast, GENE controls the variance of the effect variable, as in $R^2$, and is therefore scale invariant.

For strongly identifiable SEMs, considering the goodness of fit in an order-based manner is sufficient for causal discovery. In contrast, for weakly identifiable problems, leveraging residual independence is necessary. Therefore, after training a neural network (NN) for variable $V_{\pi(i)}$, we calculate the residual by subtracting the predicted values from the true values of $V_{\pi(i)}$. We then apply independence tests to determine whether the residual is independent of the input variables, which can be expressed as $\text{IT}(\hat{f}_i(V_{<\pi(i)}) - V_{\pi(i)}, V_{\pi(j)})$ for $j = 1, 2, \ldots, i-1$, where IT refers to the independence tests. If variables $X$ and $Y$ are independent, then $P(X, Y) = P(X) \cdot P(Y)$. For the residual and $V_{\pi(j)}$ to be tested, we discretize them into $m$ bins and calculate the contingency table. For the null hypothesis that the residual and $V_{\pi(j)}$ are independent, we apply the statistic $\chi^2$ to test it. The statistic $\chi^2$ is used to compute the $p$-value for the null hypothesis. We compare the $p$-value with the significance level of $0.05$ to determine whether the residual is independent of $V_{\pi(j)}$. The residual independence term discussed above is then added as a penalty term, yielding the final fitness function $\text{fit}(\pi)$ for an order $\pi$ as follows:

$$\text{fit}(\pi) =$$
$$\sum_{i=1}^{d} R^2\left(V_{\pi(i)}\right) \cdot \left(1 - \alpha \sum_{j=1}^{i-1} \frac{1}{i} \cdot \text{IT}\left(\hat{f}_i(V_{<\pi(i)}) - V_{\pi(i)}, V_{\pi(j)}\right)\right),$$
$$(6)$$

where IT returns 1 if the $p$-value is less than the significance level; otherwise, it returns 0. The hyper-parameter $\alpha$ is used to control the weight of the penalty term. The intuition behind this formula is straightforward: for each potential parent, if it is not independent of the residual, its contribution to predicting the regressor should be discounted. Generally speaking, this approach of combining the goodness of fit with residual independence is both neat and effective. For strongly identifiable SEMs, $R^2$ is sensitive to the direction of regression (i.e., $R^2$ differs significantly when regressing in the correct direction compared to the incorrect direction) and to the residual independence. In contrast, for weakly identifiable SEMs, the residual independence remains sensitive. Since in most cases it is unknown in advance whether a problem is strongly or weakly identifiable, GENE, as a generic approach, demonstrates its advantage over other methods that exploit only one of these aspects.

In our implementation, the hyper-parameters are set as follows: hidden layer size $h = 256$, number of discretization bins $m = 10$, $\alpha = 1$, and the significance level of the chi-square test is set to $0.05$.

## 4.2. Greedy Order Search

We now have the fitness function for a given order $\pi$. In this subsection, we describe how to search for the order with the highest fitness.

Consider starting with a random permutation (order) of magnitude $d$, denoted as $\pi^0$. A straightforward approach is to apply operations (e.g., changing the position of some elements in $\pi^0$) to modify $\pi^0$ and check if the fitness improves. If so, we save the modification and proceed to the next operation until no further improvement in fitness is possible. This process is known as greedy search in the optimization literature and is widely used in combinatorial optimization problems and causal discovery, such as Greedy Equivalence Search (GES) (Chickering, 2002). We define such an operation as follows:

**Definition 4.1** (Operation). Given a permutation of magnitude $d$, denoted as $\pi = (\pi_1, \pi_2, ..., \pi_d)$, and two integers $i$, $j$, where $i, j = 1, 2, ..., d$ and $i \neq j$. A mapping maps $\pi$ to $\hat{\pi}$ such that

$$\hat{\pi} = \begin{cases} (\pi_1, \ldots, \pi_{i-1}, \pi_{i+1}, \ldots, \pi_j, \pi_i, \pi_{j+1}, \ldots, \pi_d), & \text{if } i < j, \\ (\pi_1, \ldots, \pi_{j-1}, \pi_i, \pi_j, \ldots, \pi_{i-1}, \pi_{i+1}, \ldots, \pi_d), & \text{if } i > j, \end{cases}$$
$$(7)$$

is defined as an operation $\text{OP}_{ij}$, i.e., $\hat{\pi} = \text{OP}_{ij}(\pi)$.

With the operations defined, the greedy search process can be described by Algorithm 1 in Appendix D. We wonder how effective such a greedy optimization strategy is. Fortunately, Theorem 4.3 guarantees that, under a mild assumption, this search process can always find the global optimum, i.e., a true order $\pi \in \Pi^{\mathcal{G}}$, within a finite number of steps.

Before introducing the theorem, we define a metric to measure the discrepancy between an order and the true causal order of a DAG. Given an order $\pi$ and a DAG $\mathcal{G}$, we define the degree of reversal for $\pi$ w.r.t. $\mathcal{G}$, denoted as $Rev(\mathcal{G}, \pi)$.

**Definition 4.2** (Degree of Reverse). Given a DAG $\mathcal{G} = (V, E)$ and a permutation $\pi$ on $V$, the degree of reverse for $\pi$ on $\mathcal{G}$ i.e., $\text{Rev}(\mathcal{G}, \pi)$ is defined as:

$$\text{Rev}(\mathcal{G}, \pi) =$$
$$\left|\{(V_i, V_j) \mid \text{a path } V_i \rightarrow V_j \text{ exist in } \mathcal{G} \text{ and } V_j \text{ precedes } V_i \text{ in } \pi\}\right|.$$
$$(8)$$

In other words, $\text{Rev}(\mathcal{G}, \pi)$ is defined as the number of variable pairs that form a path in $\mathcal{G}$ but are incorrectly ordered in $\pi$. Obviously, for any $\pi \in \Pi^{\mathcal{G}}$, $\text{Rev}(\mathcal{G}, \pi) = 0$. Furthermore, for a given DAG $\mathcal{G}$, $\text{Rev}$ can be used to measure the distance between an order $\pi$ and its true order set $\Pi^{\mathcal{G}}$. The larger $\text{Rev}(\mathcal{G}, \pi)$, the less accurately $\pi$ describes the causal order. With this definition, we present Theorem 4.3.

**Theorem 4.3** (Global Optimality of Greedy Order Search). *Given an order fitness evaluation function $fit$ and a sample matrix $D$ sampled from the DAG $\mathcal{G}$, if for any orders $\pi_1$ and $\pi_2$ with $\text{Rev}(\mathcal{G}, \pi_1) < \text{Rev}(\mathcal{G}, \pi_2)$, we have $\text{fit}(D, \pi_1) > \text{fit}(D, \pi_2)$, then starting from any initial order $\pi_0$ and greedily applying operations until the fitness value cannot increase (as in Algorithm 1), the final order $\pi^*$ will be in $\Pi^{\mathcal{G}}$.*

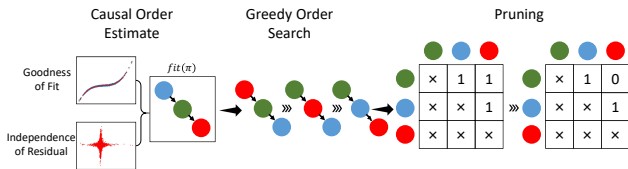

*Figure 2.* An overview of GENE. Circles with different colors stand for different variables.

**Proof Sketch:** To prove Theorem 4.3, we demonstrate that for any order $\hat{\pi}$ with non-zero $\mathrm{Rev}$, there exists an operation that can improve its $fit$. We find a pair of variables $(V_i, V_j)$ such that a path $V_i \rightarrow V_j$ exists in $\mathcal{G}$ and $V_j$ precedes $V_i$ in $\hat{\pi}$. If $V_i$ and $V_j$ are adjacent in $\hat{\pi}$, moving $V_i$ in front of $V_j$ decreases $\mathrm{Rev}$ and increases $fit$. If they are not adjacent, we consider two cases: (1) if variables between $V_j$ and $V_i$ do not form a path to $V_i$ in $\mathcal{G}$, the same operation decreases $\mathrm{Rev}$; (2) if a variable $V_t$ forms a path $V_t \rightarrow V_i$, a path $V_t \rightarrow V_j$ must exist due to the definition of a path in DAG. We recursively consider the pair $(V_t, V_j)$ until the variables are adjacent or case (1) holds. Thus, an operation exists to increase $fit$, and the greedy search will find a $\pi^*$ with $\mathrm{Rev}(\mathcal{G}, \pi^*) = 0$. The complete proof is in Appendix C.

Theorem 4.3 provides a theoretical guarantee for the global optimality of the greedy order search. Namely, it can always find the true order under mild assumptions.

### 4.3. Least Pruning

After the above two steps, we get a causal order. The remaining task is to select appropriate parents for each variable, which is essentially a variable selection problem. If we assume an additive SEM, this can be handled by Group LASSO (Ravikumar et al., 2007) or its improved version with sparsity-smoothness penalty (Meier et al., 2009). However, ANM is not additive since here additive means the effect of each parent is additive, but not additive noise. To this end, we propose a pruning strategy named least pruning which applies parent pruning to deal with such a parent selection issue. The intuition is: Starting from the given order, we construct its corresponding complete DAG, and then for each variable we try to prune one parent with least after-pruning effect. More specifically, for variable $V_{\pi(i)}$, we first calculate its fully connected $R^2$, i.e., all the variables preceding $V_{\pi(i)}$ in $\pi$, denoted as the set $\{< V_{\pi(i)}\}$ are used as input of NN. After that we try to prune each $V$ in $\{< V_{\pi(i)}\}$ and calculate $R^2$ respectively, the variable with highest after-pruning $R^2$ are called candidate since pruning it casts least effect to the overall $R^2$. We measure whether the decline of $R^2$ caused by pruning the candidate is less than a threshold, if so, prune this candidate and find the next one or else stop pruning for $V_{\pi(i)}$ and start to prune

$V_{\pi(i+1)}$. The above process is depicted in Algorithm 2 in Appendix E.

The overall framework of the least pruning is simple yet effective. There are mainly two points: How to evaluate the parent with least after-pruning effect and how to determine whether prune it or not. In this implementation we use the difference in $R^2$ before and after a pruning to measure its influence. The insight is that a variable with the least difference is with the least possibility to be the parent.

Based on the above aspects, we show the process of GENE in Figure 2. In the next section, we present experimental results on both synthetic datasets and real-world application.

## 5. Experiments

In this section, we present experimental results of performance comparison between GENE and several algorithms on synthetic datasets as well as a real-world application.

The baselines are chosen from different categories of methods: continuous-optimization-based methods include Notears-MLP (Zheng et al., 2020) and GranDAG (Lachapelle et al., 2020). For combinatorial-optimization-based methods in DAG space, we choose SELF (Cai et al., 2018), GSF (Huang et al., 2018) (Chickering, 2002), and PC-ANM, which respectively represent score-based and hybrid methods (pure constraint-based methods are not included since they can only output graphs with undirected edges). For order-based methods, R2Sort (Reisach et al., 2023), CAM (Bühlmann et al., 2014), RESIT (Peters et al., 2014), CaPS (Xu et al., 2024) and NHTS (Hiremath et al., 2024) are included.

For evaluation, we choose F1 Score and SHD (Structural Hamming Distance) as metrics. The F1 Score ranges from 0 to 1 and provides an intuitive and comprehensive assessment of the prediction quality, while the SHD is an integer that indicates the number of missing and unexpected edges. All the experiments are conducted on a computer equipped with an AMD Ryzen 5 3600 6-Core processor, 16 GB of RAM, and a 512 GB SSD. The operating system was Windows 11, 64-bit. All computational analyses were performed using Python 3.8. This setup provided a robust environment for running the extensive simulations and data processing required for the study (note that our experiments do not necessarily require GPU).

The codes and datasets involved in our experiments are available at `https://github.com/ECNU-ILOG/GENE`.

### 5.1. On Synthetic Datasets

To study the performance of our proposed GENE for problems with different features, we follow the popular experiment pipeline in the causal discovery community, which

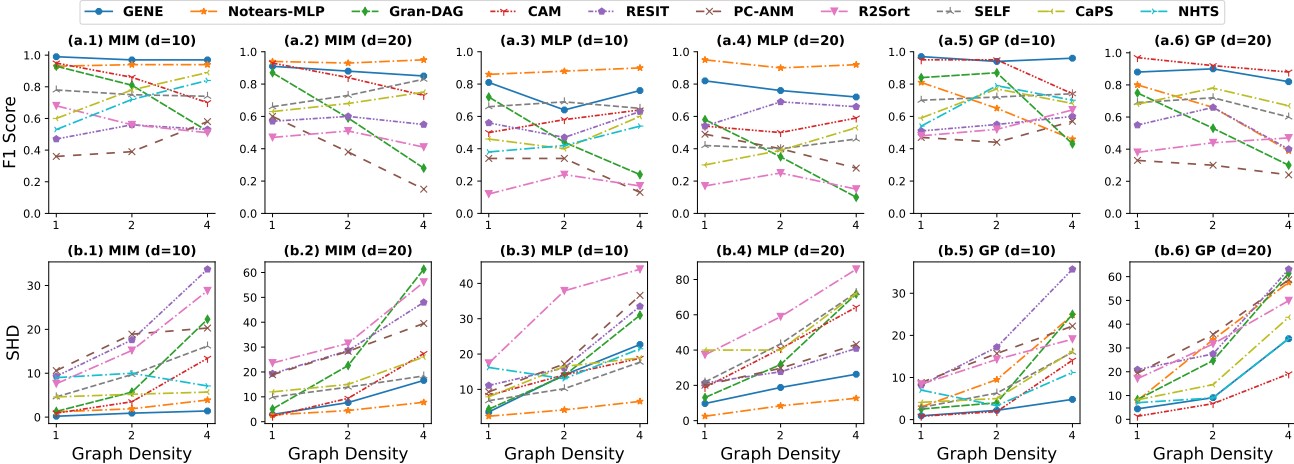

*Figure 3.* Effectiveness of the compared methods. The higher F1 the better, the lower SHD the better.

randomly generates DAGs and samples from them. We then attempt to infer the underlying DAG from the sampled data.

We evaluate the compared methods and GENE with the assumption of ANM, whose SEM is of the form Eq. (2). Specifically, we generate samples according to the following configurations: The graph sampling scheme is *Erdös Rényi* (ER), the number of nodes $d = \{10, 20\}$, the density of the graph (the number of edges divided by the number of nodes) $density = \{1, 2, 4\}$, the sample size $n = 3000$, the non-linear function $f$ has 3 forms: trigonometric function, randomized Gaussian process (GP) and randomized NN denoted as MIM, GP and MLP respectively, where MIM and GP are considered as strongly identifiable and MLP is weakly identifiable. They are given as

$$\text{MIM}: V_i = \tan(V_{\text{pa}(i)} \cdot W_1) + \sin(V_{\text{pa}(i)} \cdot W_2) +$$
$$\cos(V_{\text{pa}(i)} \cdot W_3) + c \cdot N_i,$$
$$\text{MLP}: V_i = Sigmoid(V_{\text{pa}(i)} \cdot W_1) \cdot W_2 + c \cdot N_i,$$
$$\text{GP}: V_i = GP(V_{\text{pa}(i)}) + c \cdot N_i,$$

(9)

where $W$ are random weight matrix in which the weights are calculated by $w = \pm 2^k$ where $k$ is uniformly distributed in range$(-1, 1)$ and the sign is evenly divided, $c$ is the noise coefficient which is set to 1 and $N_i \sim N(0, 1)$.

All in all, there are $2 \times 3 \times 3 = 18$ settings in total, for each setting we generate 10 different graphs and sample from them, so we get 10 sample matrix (10 problems) for each setting. All the results for one setting shown below are averaged over 10 times independent repetition on each of these 10 problems. The data generation process are implemented using the **gcastle** python toolkit (Zhang et al., 2021).

### 5.1.1. EFFECTIVENESS

The performance measured by F1 Score and SHD is given in Figure 3. If a curve for a particular baseline is missing in a graph, it indicates that the method is unable to produce results within 2 hours under the corresponding setting. From the figure, we observe that for F1 Score (shown in the upper part), GENE has advantages over other methods in terms of both F1 Score and stability to problem density in MIM and GP when $d = 10$, and in MIM when $d = 20$. However, in MLP, Notears-MLP achieves a higher F1 Score. This can be explained by information leakage, as discussed before, where its performance varies significantly after standardization. Related experiments are discussed later. The results of the performance estimated by SHD are shown in the lower part of Figure 3. From the figure, we observe that in (b.1) and (b.5), GENE achieves superior performance compared to other methods. In (b.2) and (b.4), GENE is only outperformed by Notears-MLP, whose performance is unreliable (as it drops significantly after standardization, which we will demonstrate later).

It is worth noting that methods relying solely on goodness of fit, namely CAM, Notears-MLP, R2Sort, CaPS, and NHTS, demonstrate universally and significantly better performance on strongly identifiable problems (GP and MIM) compared to weakly identifiable problems (MLP). This phenomenon further support the meaningfulness of the division between strong and weak identifiability. Besides, Figure 3 suggests the performance of GENE and CAM is very close, even in weakly identifiable cases. To address this point, we conduct an ablation experiment comparing only the quality of orders given by order-based methods in Section 5.1.5. From this, it is clear that the quality of orders given by GENE far exceeds that of CAM and RESIT. Therefore, we infer that the closeness of F1 Score and SHD is due to differences in pruning.

### 5.1.2. STANDARDIZATION

This part serves as a supplement to the effectiveness analysis discussed above. As discussed above, continuous-optimization-based methods are sensitive to data scaling. (Reisach et al., 2021) show this for the linear case, and we extend their results to the non-linear case.

Specifically, for problems with $d = 10$ and $density = 2$, we standardize the sample matrix (i.e., each variable is subtracted by its mean and divided by its standard deviation) to examine whether the performance of the above methods is influenced. The results are shown in Figure 4. From the figure, we observe that continuous-optimization-based methods, namely Notears-MLP and Gran-DAG, are greatly affected by standardization, while others remain largely unaffected. This observation aligns with our expectations. Moreover, for strongly identifiable problems, the average F1 Score decline ratio is 34%, while for weakly identifiable problems, this number is 52%. This suggests that standardization has a larger impact on weakly identifiable problems for continuous-optimization-based methods.

### 5.1.3. EFFICIENCY

To study the efficiency of the compared methods, we report the average after standardization F1 Score vs. wall-clock execution time in two-objective Pareto graphs on problems with $d = 10$ and $d = 20$, as shown in Figure 8 in Appendix F.1. The result suggests that GENE is on the Pareto front of effectiveness and efficiency, i.e., nondominated by other methods w.r.t. these two objectives.

### 5.1.4. ABLATION STUDY

Figure 5 shows the result of the ablation study for GENE where we consider the case $density = 2$. In this experiment,

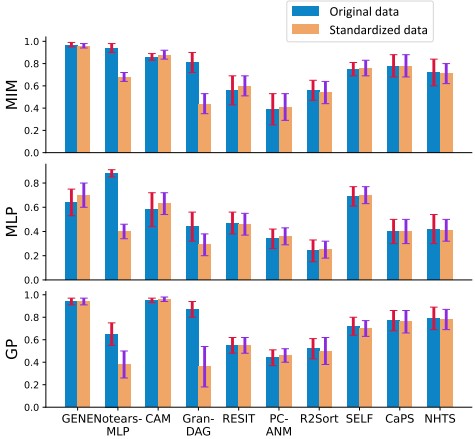

*Figure 4.* The F1 Score before and after data standardization for different causal discovery methods when $d = 10$, $density = 2$, The error bars show the standard deviation.

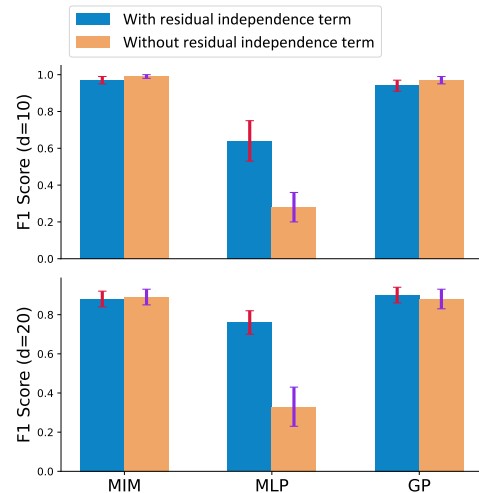

*Figure 5.* The F1 Score before and after removing the residual independence term in GENE, $d = \{10, 20\}$, $density = 2$.

we try to only exploit the goodness of fit and ignore the residual independence in order estimation, more specifically, we use $\hat{fit}(D, \pi) = \sum_{i=1}^{d} R^2\left(V_{\pi(i)}\right)$ in place of Eq. (6) and keep other settings the same. From the results we can tell that for strongly identifiable problems (GP, MIM), the performance is not influenced (results are even slightly better for GP and MIM in 10-node case) while for MLP, which is considered weakly identifiable, the performance drops significantly. This phenomenon supports the discussions about the strength of identifiability.

### 5.1.5. EFFECTIVENESS OF ORDER SEARCH

The performance of order-search-based methods (namely CAM, RESIT, CaPS, NHTS and GENE) are not only affected by the quality of obtained order, but also by the parent search (pruning) process. Thus to study the effectiveness of order search, i.e., only consider the quality of orders obtained by these methods, we drop the pruning process and only compare the ability to search for correct causal orders of these methods, which can be evaluated by the degree of reverse Rev (defined in Section 4.2) of the found order. Specifically, we conduct experiments for order-search-based methods on problems with the graph sampling scheme ER, the number of nodes $d = 10$ and $density = \{1, 2, 4\}$, the sample size $n = 3000$ and the non-linear form MIM, GP and MLP. and the output is an order rather than a DAG, for which we calculate its degree of reverse Rev as in Eq. (8). The result is shown in Figure 6.

From Figure 6, we can see it more clearly about the power of Eq. (6): In (b) which is considered as weakly identifiable, GENE performs apparently better than other compared methods (its Rev is almost a half of that of CAM and a quarter of RESIT) and in (a) and (c) which are considered as

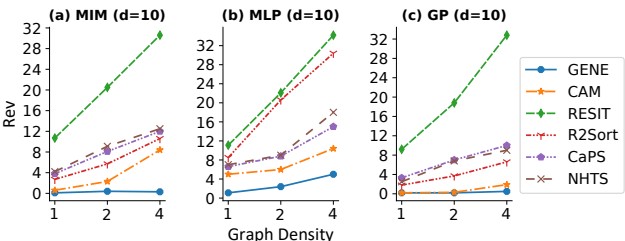

*Figure 6.* The average degree of reverse for output order by GENE, CAM and RESIT. (a) MIM with d = 10. (b) MLP with d = 10. (c) GP with d = 10. The lower Rev the better.

strongly identifiable, GENE can find orders with Rev nearly equals to 0, that means the order found by GENE is very close to the true order. It is worth noting that even in strongly identifiable case ((a) and (c)), GENE performs better than CAM especially when $density = 4$, it means that even for strongly identifiable problems the residual independence term also provide some help to find the true order, this is what we do not expected before.

For more experimental results and analysis, including the results of efficiency and analysis of hyper-parameter influence, please refer to Appendix F. In the next subsection, we will verify how the proposed GENE performs in a real-world application with bioinformatics background.

### 5.2. On Real-World Application

Causal discovery has significant and far-reaching applications in causal protein signaling networks (Sachs et al., 2005). Pathways in protein signaling networks can be simply understood as a series of enzyme-catalyzed reaction pathways through which molecular signals are transmitted from outside the cell through the cell membrane to the inside of the cell to produce an effect. The various biochemical reaction pathways that perform different functions in the cell are composed of a series of different proteins, and the regulation of the activation or inhibition state of different proteins is mainly achieved by adding or removing phosphate groups. The above problems can be solved with the idea of causal graphical models (Sachs et al., 2005), which illustrates the causal relations between the components of the pathway in the form of a DAG-represented causal graph.

We consider a popular causal protein signaling network dataset in causal discovery community named Sachs (Sachs et al., 2005). Sachs contains 7466 observational and interventional samples of a protein signaling network based on expression levels of proteins and phospholipids. The ground truth causal graph is labeled by domain experts and it has 11 nodes and 17 edges (Sachs et al., 2005). Sachs is widely used in the community of causal discovery, and it is challenging for the most causal discovery methods. Herein, we use the 853 observational data to conduct causal discovery.

*Table 1.* Results on Sachs (bold means the best).

| Method | F1 Score | SHD |
|---|---|---|
| GENE | **0.65** | **11** |
| Notears-MLP | 0.24 | 19 |
| Gran-DAG | 0.33 | 16 |
| CAM | 0.38 | 16 |
| RESIT | 0.15 | 23 |
| PC-ANM | 0.1 | 18 |
| GSF | 0.23 | 20 |
| R2Sort | 0.35 | 17 |
| CaPS | 0.07 | 17 |
| NHTS | 0.17 | 19 |

The results are shown in Table 1. From which, GENE remarkably outperforms other methods on both F1 Score and SHD. The comparison between the causal graphs discovered by GENE and the ground truth is in Figure 7.

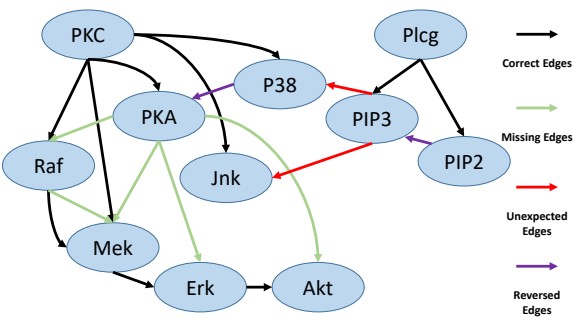

*Figure 7.* The protein signaling graph discovered by GENE.

## 6. Conclusion & Discussion

This paper focuses on the additive noise causal discovery problems and proposes to divide identifiability into strong and weak ones. The existence of implicit functions making a large difference on the difficulty of a causal discovery problem, and the existing methods mainly perform well on causal models with strong identifiability. We show that this failure is due to the ignorance of residual independence. Based on this observation, we propose GENE, a unified and generic approach for both the strong and the weak ones. GENE takes the residual independence into account and thus is able to deal with both types of problems.

For the future work, the performance of pruning strategy in GENE can be further enhanced. Besides, the strength of identifiability can be considered and generalized under other assumptions and scenarios, e.g., the post-nonlinear case.

## Acknowledgements

The authors would like to express the sincere thanks to the anonymous reviewers for their constructive comments and suggestions. This work is supported by the National Natural Science Foundation of China (No. 62476091) and National Postdoctoral Program for Innovative Talent (BX20240162).

## Impact Statement

This paper presents work whose goal is to advance the field of Machine Learning. There are many potential societal consequences of our work, none which we feel must be specifically highlighted here.

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

# Appendix

# A. Detailed Related Work

From the perspective of optimization, the existing causal discovery methods can be divided into two types: combinatorial-optimization-based methods and continuous-optimization-based methods. The task of causal discovery is naturally a combinatorial optimization problem since the search space of DAG is discrete. However, Zheng et al. (2018) present another possibility for causal discovery. They reformulate the problem in continuous space via transforming the discrete objective function and constraint into continuous and differentiable ones. In this section, we briefly introduce these two classes of causal discovery methods respectively.

## A.1. Combinatorial-Optimization-based Methods

The combinatorial-optimization-based methods address the causal discovery problem via searching for a graph represented by a connectivity matrix in the DAG space. They can be further classified by its search space, i.e., the DAG space and the order space. For methods searching in the DAG space, there are three categories: constraint-based, score-based and hybrid methods.

Constraint-based methods apply independent tests and conditional independence tests to infer the independence property between variables. Particularly, when considering non-linear problems, kernel-based independence tests are often applied, such as (Zhang et al., 2011). Representative constraint-based algorithms include Peter-Clark algorithm (PC) (Spirtes et al., 2000), inductive causation algorithm (IC) (Verma & Pearl, 1990). The main drawback of this kind of methods is that they cannot identify graphs in the same MEC, and thus there may exist undirected edges in the output graph. Besides, constraint-based methods also suffer from a series of problems of conditional independence tests including conflict handing, sample inefficiency, and hyper-parameter sensitivity, etc.

The second category is score-based methods, which first define a score function using to evaluate the fitness degree of the input DAG and the observational data. For non-linear causal discovery task, the generalized score function (GSF) (Huang et al., 2018) and the structural equational likelihood framework (SELF) (Cai et al., 2018) are proposed. SELF takes the both two aspects into account. However, SELF does not discuss the relationships between the two aspects and identifiability. After defining the score function, score-based methods then conduct a searching process to find the DAG that maximizes the score. The popular search strategies include hill climbing (HC), greedy equivalence search (GES) (Chickering, 2002). The defect of this kind of methods is that they formulate the whole problem

as an optimization problem and do not explicitly discuss the identifiability of the output graph. Besides, due to the use of gradient-free optimization to search for structures across the entire DAG space, score-based methods often have significant efficiency shortcomings.

The hybrid methods first apply constraint-based methods to identify the V-structure. The remaining undirected edges are seen as cause-effect pairs and the direction is determined following those identifiability results such as ANM aforementioned in the introduction (details will be introduced in Section 2.2). Representative algorithms include PC-ANM (Hoyer et al., 2008), scalable causation discovery algorithm (SADA) (Cai et al., 2013). However, due to the aforementioned drawbacks of conditional independence tests, especially kernel-based independence tests for non-linear problems, the performance of constraint-based methods is not stable, which indeed infects the effectiveness of hybrid methods.

Except for searching in a DAG space, an alternative is searching in an order space. The proposed GENE belongs to this class. In the order-search-based non-linear causal discovery, representative algorithms include causal additive model (CAM) (Bühlmann et al., 2014), regression with subsequent independence test algorithm (RESIT) (Peters et al., 2014), causal discovery with parent score (CaPS) (Xu et al., 2024) and causal discovery for non-linear ANMs with local ancestor-descendent relationships (NHTS) (Hiremath et al., 2024).

On the one hand, CAM applies a log-likelihood-based measurement which captures the goodness of fit to estimate an order. With such measurement, CAM just greedily adds edges to maximize this log-likelihood until getting a complete directed graph, which yields an order as the result. NHTS (Hiremath et al., 2024) propose a hybrid approach that first establishes a top-down hierarchical ordering leveraging ancestral relationships, applicable to both linear and nonlinear additive noise models, and subsequently employs a nonparametric constraint-based algorithm with local search for pruning spurious edges. Similarly, CaPS (Xu et al., 2024) is presented as an ordering-based algorithm specifically designed to handle a mixture of linear and non-linear relations. CaPS introduces a novel identification criterion for topological ordering and incorporates a parent score, quantifying the strength of average causal effects, to optimize its pruning stage. Unfortunately, these methods only consider the goodness of fit, and thus they can only handle strongly identifiable problems.

On the other hand, RESIT considers the problem from another perspective. For each node $V_i$, the corresponding noise term $N_i$ is independent from all non-descendants of $V_i$. Thus for the leaf node $V_{leaf}$, its noise $N_{leaf}$ is independent of all other variables. Based on the above obser-

vation, RESIT proposes to iteratively pick one node with least dependency between other variables and its residual regressing on all other variables. This process undoubtedly yields an order (from leaf to root). This leaf-node-based order search method is also applied by recently proposed score matching causal discovery algorithms (Rolland et al., 2022; Sanchez et al., 2023). However, RESIT does not work well for strongly identifiable problems since it ignores the information from the goodness of fit.

The existing order-search-based approaches only consider one of the two aspects to estimate the causal order, and both of them have limitations. This motivates the work in this paper, i.e., unifying these two aspects under one framework and proposing a generic method for both strongly and weakly identifiable problems.

### A.2. Continuous-Optimization-based Methods

Unlike combinatorial methods which solve the problem in discrete spaces, Zheng et al. (2018) propose to conduct causal discovery via optimizing a mean square error (MSE) based loss function subject to an equivalent continuous acyclicity constraint. This idea achieves a remarkable performance on popular benchmarks and many successful extensions have been proposed including (Lachapelle et al., 2020; Bhattacharya et al., 2021; Pamfil et al., 2020; Yu et al., 2019; Brouillard et al., 2020; Ng et al., 2020; Wei et al., 2020; Lee et al., 2020; Zheng et al., 2020). Vowels et al. (2021) provide a comprehensive review of these continuous-optimization-based methods.

However, Reisach et al. (2021) point out that these continuous-optimization-based methods may exploit information inadvertently leaked by the data generating process to achieve such extraordinary performance. Reisach et al. (2021) show that under the linear model with additive noise case, the performance of continuous-optimization-based methods drop significantly after data rescaling, i.e., changing the units of measurement for variables (e.g., multiplying 1000 from kilometer to meter) or data standardization. This result also holds under non-linear settings (Reisach et al., 2023), which is also shown in our experiment section. Apparently the existence of causal relationships should not depend on the scale of data. This phenomenon is because these continuous-optimization-based methods treat the DAG connectivity matrix as a whole during optimization, which means that they do not explicitly distinguish between the cause variables and the effect variable. However, in GENE the cause variables and the effect variable are clear when conducting regression. Hence GENE is scale invariant. Besides, these methods are also lack of discussion about identifiability.

## B. Proof of Lemma 3.2

*Proof.* Consider $T \colon C[a, b]^n \to C[a, b]^n$ defined by

$$(T\varphi)(\boldsymbol{x}) = \varphi(\boldsymbol{x}) - \frac{1}{M} F(\boldsymbol{x}, \varphi(\boldsymbol{x})),$$
$$\text{for all } \boldsymbol{x} \in [a, b]^n \text{ and } \varphi \in C[a, b]^n.$$

$T$ is a contraction on $C[a, b]^n$ since

$$
\begin{aligned}
&\left| (T\varphi_2)(\boldsymbol{x}) - (T\varphi_1)(\boldsymbol{x}) \right| \\
&= \left| \varphi_2(\boldsymbol{x}) - \tfrac{1}{M} F(\boldsymbol{x}, \varphi_2(\boldsymbol{x})) - \varphi_1(\boldsymbol{x}) + \tfrac{1}{M} F(\boldsymbol{x}, \varphi_1(\boldsymbol{x})) \right| \\
&= \left| \left( 1 - \tfrac{1}{M} F_y\big(\boldsymbol{x}, \varphi_1(\boldsymbol{x}) + \theta(\varphi_2(\boldsymbol{x}) - \varphi_1(\boldsymbol{x}))\big) \right) \right. \\
&\quad \left. \cdot (\varphi_2(\boldsymbol{x}) - \varphi_1(\boldsymbol{x})) \right| \\
&\leq (1 - \tfrac{m}{M}) \left| \varphi_2(\boldsymbol{x}) - \varphi_1(\boldsymbol{x}) \right|,
\end{aligned}
$$

where the second equation holds because of the Lagrange mean value theorem (Rudin et al., 1976). The fact that $T$ is a contraction implies

$$dis(T\varphi_2, T\varphi_1) \leq \alpha \cdot dis(\varphi_2, \varphi_1),$$

where $dis$ is a metric on space $C[a, b]^n$ and $\alpha \in (0, 1)$. Hence, by the Banach fixed point theorem (Rudin et al., 1976), there exists a unique $\varphi \in C[a, b]^n$ that $F(\boldsymbol{x}, \varphi(\boldsymbol{x})) \equiv 0$ on $[a, b]^n$, and the lemma holds. □

## C. Proof of Theorem 4.3

*Proof.* Theorem 4.3 states that, under the assumption that if an order is with lower Rev, it is with higher $fit$, then a greedy order search procedure will result in a correct causal order. To prove this, it is equivalent to prove the following statement. For any $\pi$ with non-zero Rev, there exist some operations can increase its $fit$. If this statement holds, obviously the greedy search will finally find a $\pi^*$ with $\mathrm{Rev}(\mathcal{G}, \pi^*) = 0$, i.e., $\pi^* \in \Pi^{\mathcal{G}}$. Without loss of generality, we consider a $\hat{\pi}$ that $\mathrm{Rev}(\mathcal{G}, \hat{\pi}) > 0$. From the definition of Rev, we can at least find a pair of variables $(V_i, V_j)$ such that a path $V_i \to V_j$ exists in $\mathcal{G}$ and $V_j$ precedes $V_i$ in $\hat{\pi}$. If there are no variables between $V_i$ and $V_j$, i.e., they are adjacent in $\hat{\pi}$, then the operation that moves $V_i$ in front of $V_j$ can make Rev minus 1 and $fit$ increases. This means that such an operation exists, and thus the statement holds.

Else if $V_i$ and $V_j$ are not adjacent in $\hat{\pi}$, we still consider the above operation, i.e., moving $V_i$ in front of $V_j$. Now there are two cases. For the first case, if all variables between $V_j$ and $V_i$ do not form a path to $V_i$ in $\mathcal{G}$, then such an operation still makes Rev minus 1. For the second case, if there exists a variable $V_t$ forming a path $V_t \to V_i$ in $\mathcal{G}$, then such an operation no longer decreases Rev since it introduces a new pair $(V_i, V_t)$ which is reversed after the operation. However, from the definition of path in DAG, we know that if a path

$V_i \to V_j$ and another path $V_t \to V_i$ both exist, then a path $V_t \to V_j$ exists. So we can consider the pair $(V_t, V_j)$ just like considering the pair $(V_i, V_j)$ before. This is a recursive process until we find that the pair of variables are adjacent in $\hat{\pi}$ or the pair of variables are not adjacent but variables between them do not form path to the latter one (as the first case). Therefore the operation exists under both of the two circumstances, and the theorem holds. □

## D. Greedy Order Search

The pseudocode of the greedy search algorithm described in the main text 'Greedy Order Search' section is shown in Algorithm 1.

---

**Algorithm 1** Greedy Search for Order

---

**Input:** the sample matrix $D \in \mathbb{R}^{n \times d}$, the fitness evaluation function $fit$.

1: initialize the current best order $\pi^* \leftarrow Random\ Order$, the best fitness value $bestfit \leftarrow \text{fit}(D, \pi^*)$, a Boolean value $continue \leftarrow True$
2: **while** $continue = True$ **do**
3:    $continue \leftarrow False$
4:    **for** $i = 1 \to d$ **do**
5:      **for** $j = 1 \to d, j \neq i$ **do**
6:        $\hat{\pi}^* = \text{OP}_{ij}(\pi^*)$
7:        **if** $\text{fit}(D, \hat{\pi}^*) > bestfit$ **then**
8:          $\pi^* \leftarrow \hat{\pi}^*, bestfit \leftarrow \text{fit}(D, \hat{\pi}^*)$
9:          $continue \leftarrow True$
10:        **end if**
11:      **end for**
12:    **end for**
13: **end while**
14: **Output:** the order found with the highest fitness $\pi^*$

---

## E. Least Pruning

The pseudocode of the least pruning algorithm described in the main text 'Least Pruning' section is shown in Algorithm 2.

## F. Additional Experimental Results

### F.1. Efficiency

The results of efficiency are shown in Figure 8, the related analysis is available in Section 5.1 **Efficiency** part.

### F.2. Hyper-parameter Analysis

We investigate the influence of hyper-parameters, namely $\alpha$ in Eq. (6) which is used to control the ratio of the penalty term given by residual independence. Specifically, we set

---

**Algorithm 2** Least Pruning

---

**Input:** the sample matrix $D \in \mathbb{R}^{n \times d}$, an order $\pi$, and a threshold $\theta$

1: $\mathcal{G} \leftarrow \emptyset$
2: **for** $i = 1 \to d$ **do**
3:    the parent set $Pa \leftarrow V_{<\pi(i)}$
4:    regress $V_{\pi(i)}$ over $V_{<\pi(i)}$ and calculate $R^2$, $fit \leftarrow R^2$
5:    **while** $|Pa| > 0$ **do**
6:      **for** $j = 1 \to i$ **do**
7:        remove $V_{\pi(j)}$ from $Pa$ and calculate $R_j^2$
8:      **end for**
9:      find the parent $V_{least}$ with the least $fit - R_{least}^2$
10:      **if** $fit - R_{least}^2 < \theta$ **then**
11:        prune it, $Pa \leftarrow Pa \setminus V_{\pi(least)}$, $fit \leftarrow fit - R_{least}^2$
12:      **else**
13:        break While
14:      **end if**
15:    **end while**
16:    for each parent $p$ left in $Pa$, add edge $p \to V_i$ to $\mathcal{G}$
17: **end for**
18: **Output:** the causal graph $\mathcal{G}$

---

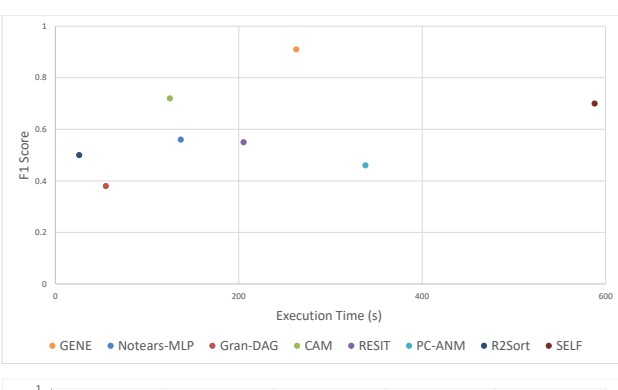

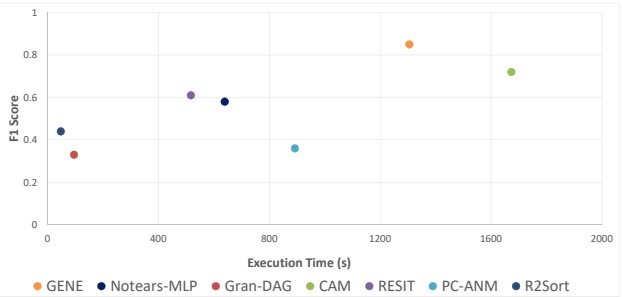

*Figure 8.* The wall-clock time vs. after standardization F1 score when $d = 10$ and $d = 20$.

$\alpha = 1$ as default in previous experiments, here we change it to $\{0.2, 0.5, 2, 5\}$ and report the performance on datasets generated by MLP, MIM and GP with $d = 10, density = 2$. The results are shown in Figure 9.

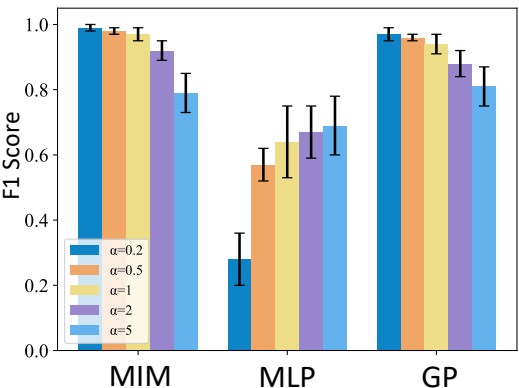

*Figure 9.* The performance of GENE with different values of hyper-parameter $\alpha$ when $d = 10$, $density = 2$.

We can observe from Figure 9 that for MIM and GP, which are strong identifiable problems, the smaller $\alpha$ seems to perform better and when $\alpha = 0.2, 0.5, 1$, the performance is close. For MLP, the weakly identifiable one, too small $\alpha$ (e.g., $0.1$) may lead a poor performance since the residual independence term plays an essential role for weak problems. Generally speaking, GENE is not sensitive to the hyper-parameter $\alpha$ in a proper range (from $0.5$ to $2$) and we recommend to set $\alpha$ in this range.

