# OpenReview forum: "Strong and Weak Identifiability of Optimization-based Causal Discovery in Non-linear Additive Noise Models"
_ICML.cc/2025/Conference — ICML 2025 poster_

### Official Review · Reviewer_isnT · 2025-03-06

**Overall Recommendation:** 2

**Summary:**

The manuscript introduces a criterion for strong vs. weak identifiability in causal graphs and explores the implications for optimization based structure discovery algorithms. Specifically, the authors propose a gradient-based approach whose objective combines a standard goodness of fit measure ($R^2$) with a residual independence test to score candidate orderings. Experiments demonstrate strong performance compared to additive noise model causal discovery algorithms on a range of synthetic and real-world benchmarks.

## update after rebuttal

I thank the authors for their rebuttal. After reading the discussion with other reviewers, I am inclined to agree that this manuscript is somewhat under-developed at present and could benefit from further experiments and/or theoretical analysis. I will be revising my score downward for consensus but encourage the authors to revise and resubmit in the near future. This paper is nearly there and will find a good home soon!

**Claims And Evidence:**

The main theoretical claim is the purported distinction between "strong" and "weak" identifiability of causal structures in additive noise models (ANMs). If I understand correctly, the structural equation for variable $V_i$ is "strongly identifiable" if we can uniquely solve for each of its parents by fixing the value of $V_i$ and all other parents (including an exogenous noise variable). So far so good. But I'm a bit confused about if/how these notions map onto classical distinctions between identifiable vs. partially identifiable structures. If an ANM is only "weakly" identifiable, does this mean that there is no unique solution (at least without further assumptions)? It's not obvious to me that this follows. If there is no unique solution, can we at least characterize the space of possible solutions (e.g., something akin to a Markov equivalence class in constraint-based approaches?) Also, the text appears to suggest that strong and weak identifiability form a partition on the space of ANMs. But surely some ANMs are simply _unidentifiable_?

The empirical results are impressive. The idea of adding a residual independence testing component to the standard objective for optimization-based causal discovery makes good sense and appears quite effective. I am unaware of any previous proposals along this line.

**Essential References Not Discussed:**

I am not aware of any essential references that were not discussed, but I confess I am not an expert in this domain.

**Experimental Designs Or Analyses:**

As noted above, the experiments are clear and compelling.

**Methods And Evaluation Criteria:**

The experiments are convincing and well-designed. Synthetic and real-world results tell a similar story.

**Other Comments Or Suggestions:**

-The $R^2$ formula is missing a $1–$ before the ratio

-Is 2 hours a reasonable time limit for causal discovery algorithms? Seems a little conservative.

-I understand the motivation for the residual independence test, but I was a bit baffled by the choice to implement this via a $\chi^2$ test. Surely we lose information by discretizing? This also introduces hyperparameters that may influence results (how many bins to use?) There are plenty of nonparametric independence tests for continuous data that could be used instead, e.g. Spearman $\rho$ or HSIC.

**Other Strengths And Weaknesses:**

The idea is clear. The results are compelling. I feel I might be missing some link between the theoretical and empirical results. Greater elaboration on the former could help readers better appreciate the latter.

**Questions For Authors:**

My main questions are:

(1) What exactly does "weak identifiability" amount to? Is it just that we need more samples to get the right answer, or that even in the infinite limit we will not converge on a unique answer?

(2) We have a sufficient condition for weak identifiability, but none for unidentifiability (or for that matter strong identifiability). Any idea what these might look like? How about necessary conditions for any of these?

**Relation To Broader Scientific Literature:**

The topic of optimization-based causal discovery is of great interest to the ICML community, and has broad scientific application. Leveraging neural networks and gradient-based learning for this difficult task, which has traditionally been formulated with discrete reasoning, is a promising direction. Early works in this area faced some challenges (var-sortability, etc.) but I believe there will continue to be more interesting developments in this space. The present work makes a small but meaningful contribution to this discourse.

**Theoretical Claims:**

The proofs appear sound, though I did not check them closely. I have some questions about how to interpret Lemma 3.2. Why do the constants $m, M$ have to be positive? This seems like a very restrictive assumption. If I understand correctly, it means that $F$ is strictly monotone in each of its $n+1$ arguments, at least within the range $E$? Also, is $E$ meant to be the full support or just a subset? If the former, then the lemma should probably say so. If the latter, then what happens outside this range?

I'm not entirely clear to me what to make of Thm. 3.3 on its own – do we not have any sufficient conditions for strong identifiability? Or for that matter unidentifiability?

Thm. 4.3 is a nifty result. Always nice when greedy methods are globally optimal!

---

> ### Author Rebuttal · Authors · 2025-04-01
>
> We sincerely appreciate your constructive feedback and insightful suggestions. Below, we address each point raised in the review.
> ### **Claims and Evidence**
> 1. **confusion about the strength of identifiability**
> Thank you for raising this important point. Within the ANM framework—which is inherently identifiable—our distinction between strong and weak identifiability refines the classical notion of identifiability into subclasses based on practical difficulty:
> -Strong Identifiability: The causal direction is uniquely recoverable via regression alone (examples see Fig. 1), requiring no additional criteria.
> -Weak Identifiability: The causal direction is still uniquely identifiable but requires residual independence tests to resolve ambiguity.
>
> Both classes are fully identifiable under ANM assumptions, differing only in the difficulty of identification. Weak cases do not imply partial identifiability but instead demand stricter criteria to isolate the true graph. This distinction guides algorithm design: GENE adaptively combines criteria to address both regimes, ensuring robustness. We will clarify these nuances in the revision.
>
> ### **Theoretical Claims**
> 1. **Questions about Lemma 3.2**
>
> Thank you for your careful reading. m and M have to be positive because this property is used in the process of constructing the inequality proof for the contraction. The positivity ensures strict monotonicity of F, guaranteeing a unique implicit function φ(x) over the domain E. Regarding the support range of E, it is necessary to understand it in conjunction with the role of this Lemma and Theorem 3.3. The purpose of providing this theorem is to move from a purely theoretical definition of strong and weak identifiability to providing a practically operational way to detect the strong and weak identifiability of SEMs (although in causal discovery, this is rarely done).  It bridges theoretical definitions to actionable insights: if F satisfies the lemma's criteria within any plausible domain E (e.g., observed data ranges), the SEM is deemed weakly identifiable. This aligns with causal discovery's practical focus, where identifiability is assessed within empirically relevant regimes. We will clarify this intent in the revision.
>
> 2. **Confusion of Thm. 3.3**
>
> Theorem 3.3 focuses on weak identifiability because it represents the more challenging and nuanced regime where classical methods fail. Strong identifiability, by definition, arises when no implicit functions exist (Def. 3.1), making it straightforward to identify causal directions via regression alone. Thus, strong identifiability does not require separate conditions—it is the default when Theorem 3.3's criteria are unmet. Unidentifiable SEMs, fall outside our scope, as ANMs inherently assume identifibility. We will clarify this hierarchy in the revision.
>
> ### **Other Comments Or Suggestions**
> 1. **Issue of R^2 Formular**
>
> Thank you for catching this important technical detail. You are absolutely correct that we miss a $1-$ in the formulation. We sincerely appreciate your careful reading, and we will make sure to correct this in the revised manuscript.
>
> 2. **Time Limit**
>
> Since in our simulation experiments, for each setting (node_num, density, function form), we generate 10 graphs, which is equivalent to having 10 problems. Each algorithm is then repeated 10 times on each of these problems, meaning each algorithm has 10 * 10 = 100 runs for each setting. Therefore, setting the time limit to 2 hours is quite reasonable.
>
> 3. **Choice of Independence Tests**
>
> We appreciate this suggestion. While nonparametric tests (e.g., HSIC, Spearman) avoid discretization, they introduce computational bottlenecks—HSIC scales as O(n^2) per test, and Spearman requires rank calculations across O(d^2) variable pairs. The chi^2 test balances efficiency (O(n) per test) with empirical reliability. We validated binning choices (m=10) across synthetic datasets, observing stable performance. That said, we agree that discretization loses information and will explore hybrid strategies (e.g., kernel-based tests for critical nodes) in future work.
>
> ### **Questions For Authors**
> 1. **Meaning of Weak Identifiability**
>
> Weak identifiability does not imply unidentifiability. Under ANM assumptions, even weakly identifiable SEMs are uniquely identifiable in the infinite-sample limit. The distinction lies in the practical requirements: weakly identifiable cases demand residual independence tests to resolve directionality ambiguities that persist with finite data (e.g., symmetric regression fits). While strong identifiability allows causal discovery via regression alone, weak identifiability requires additional criteria—but both guarantee convergence to the true graph asymptotically.
>
> 2. **Condition for Unidentifiability or Strong identifiability**
>
> Please refers to above response for Confusion of Thm. 3.3.

---

### Official Review · Reviewer_XUn3 · 2025-03-07

**Overall Recommendation:** 2

**Summary:**

The paper tackles the problem of causal discovery in additive noise models. It identifies different classes of ANMs, strongly identifiable or weakly identifiable, that pose different levels of difficulty for traditional discovery algorithms. It characterizes sufficient conditions for the different classes and proposes a novel algorithm (GENE) that has consistent performance across both classes of ANMs. The algorithm proposes to find a topological ordering that maximizes a goodness-of-fit measure based on R^2 values and residual independence via a greedy search approach; a novel edge pruning approach that leverages R^2 values (least pruning) is proposed to enable parent set identification. The authors validate GENE real and synthetic data, finding that their algorithm maintains performance on strongly identifiable datasets while having superior performance on both weakly identifiable ANMs and real-world data.

**Claims And Evidence:**

1. On page 3, following Definition 3.1, the authors claim that if an implicit function does not exist, the SEM can be identified by simple regression - however, this is never shown formally. This is important, as the validity of the distinction between weak and strongly identifiable ANMs hinges on whether such a simplification can be made.

2. In Section 4.3, the authors introduce "least-pruning" as a way to prune spurious edges, given a correct topological ordering. They suggest that Group LASSO is inappropriate given that the ANM may not have an additive contribution from each parent. However, it is unclear how their Least Pruning approach overcomes such a limitation, or in general, what the advantage is of this approach, obscuring the paper's contribution.

3. On page 4, paragraph 4, the authors claim that "For strongly identifiable SEMs, consider the goodness of fit in an order-based manner is sufficient for causal discovery". However, this claim lacks substantiation, which casts doubt on whether the objective function proposed by the authors is well-motivated.

**Essential References Not Discussed:**

1. In the ordering stage of GENE, the authors propose to use a combination of R^2 and residual independence to define the fitness function, with greedy search as an optimization algorithm; however, this appears to be a fusion of the R^2-sortability approach discussed in [1], as well as the independence-based score approach discussed in Section 4.2 and 4.2.2 of [2] - how does the proposed approach differ from these concepts?

2. In the edge pruning stage of GENE, the suggested least pruning algorithm is extremely similar to the pruning algorithm suggested by RESIT [2] (Section 4.1), with the main difference being that the R^2 measure replaces the residual independence measure. However, this remains undiscussed in the paper, and thus the novelty of this approach appears low.



[1] Reisach, A. G., Tami, M., Seiler, C., Chambaz, A., & Weichwald, S. A scale-invariant sorting criterion to find a causal order in additive noise models. *Proceedings of Machine Learning Research*, vol TBD:1–24, 2023.

[2] Peters et. al, Causal Discovery with Continuous Additive Noise Models, (2014).

**Experimental Designs Or Analyses:**

The experimental designs presented are both sound and valid. However, there are significant additional experiments to run in order to fully evaluate the proposed approach (see 'Questions for Authors' for more details).

**Methods And Evaluation Criteria:**

The evaluation criteria (F1 and SHD) as well as the synthetic benchmarks used make sense for the application of causal discovery. However, there are significant additional experiments to run in order to fully evaluate the proposed approach (see 'Questions for Authors' for more details).

**Other Comments Or Suggestions:**

NA

**Other Strengths And Weaknesses:**

Weakness:

1. The paper's novel contribution can be considered relatively minor. The algorithm GENE simply combines existing ideas from RESIT, GES, and R^2-Sort to enable both its ordering and pruning algorithms. Although the identification of strong/weak identifiable ANMs is novel and interesting, it is underexplored in this work, with little formal reasoning about how different algorithms may succeed or fail in different types of ANMs.

**Questions For Authors:**

1. What is the intuition behind what makes strongly or weakly identifiable ANMs easier or harder to discover? Why is it that "simple regression" suffices in the strongly identifiable cases, whereas it does not in the weakly identifiable? Further, is it possible to prove how various classic methods might fail in each scenario? Without extensive explicit and formal characterization of the importance of identifiability type, the contribution of the paper remains unclear.
   1. To this end, perhaps there is a connection between invertibility and strong/weak identifiability?
2. Although the authors provide a correctness result for their ordering algorithm, there is no such correctness result for their edge pruning algorithm (least pruning). In fact, there is little discussion of the assumptions under which least pruning would be expected to be accurate, and it seems as if the approach simply leverages the R^2 heuristic. Without guarantees of correctness, it is unclear when we might expect least pruning to succeed, or outperform other approaches.
3. The experimental results can be considered an incomplete evaluation, and raise a few questions that cast doubt on the validity of the author's approach:
   1. To properly evaluate the effectiveness of the proposed edge pruning "least pruning", an ablation study must be conducted. In particular, given a correct topological ordering, the authors should compare the accuracy of the adjacency matrix yielded against other baseline methods (Lasso, CAM-pruning, Edge Discovery; see [1] for details on such an experimental setup).
   2. To further understand the performance increase over baselines, GENE should be compared against classic baselines such as DirectLiNGAM [2], as well as newer state-of-the-art approaches such as NHTS, SCORE, NoGAM, CaPS, and NHTS [1, 3, 4, 5]. Additionally, experiments should be conducted with non-gaussian noise (laplacian, uniform) to demonstrate the robustness of GENE.
   3. The causal mechanisms considered by the author contain mechanisms that either all have implicit functions, or all do not - this may not be representative of real-life scenarios, where one is likely to encounter a range of mechanisms for which an implicit function may or may not exist for each parent. The authors should design an experiment to test whether GENE still outperforms when some causal mechanisms have implicit functions, and some do not, in order to ensure their experimental results do not overstate the performance of GENE.
   4. R2-Sort performs poorly for both MIM and GP, which are both strongly identifiable ANMs - however, the authors claim in Section 4 that for strongly identifiable ANMs, considering the good-of-fit measure yielded from R^2 is sufficient for accurate discovery. This seems contradictory and casts doubt on the authors' explanation for the overperformance of GENE.



[1] Hiremath et. al, Hybrid Top-Down Global Causal Discovery with Local Search for Linear and Nonlinear Additive Noise Models, NeurIPS (2024).

[2] Shimizu et. al, DirectLiNGAM: A Direct Method for Learning a Linear Non-Gaussian Structural Equation Model, (2011).

[3] Montagna et. al, Scalable Causal Discovery with Score Matching, CLeaR 2023.

[4] Montagna et. al Causal Discovery with Score Matching on Additive Noise Models with Arbitrary Noise. PMLR 2023.

[5] Xu et. al, Ordering-Based Causal Discovery for Linear and Nonlinear Relations, NeurIPS 2024.

**Relation To Broader Scientific Literature:**

Prior work ([1,2]) has highlighted how ANMs may be generated with different characteristics (var-sortability, R^2-sortability), and propose heuristic algorithms (Var-Sort, R^2 Sort) that can exploit these characteristics to recover the underlying DAG. This paper outlines a novel characteristic of ANMs, strong or weak identifiability, and constructs a heuristic algorithm to exploit recovery in these contexts.

[1] Reisach, A. G., Seiler, C., & Weichwald, S. Beware of the simulated DAG! Causal discovery benchmarks may be easy to game. *Proceedings of Machine Learning Research*, vol TBD:1–24, 2021.

[2] Reisach, A. G., Tami, M., Seiler, C., Chambaz, A., & Weichwald, S. A scale-invariant sorting criterion to find a causal order in additive noise models. *Proceedings of Machine Learning Research*, vol TBD:1–24, 2023.

**Theoretical Claims:**

I checked the proof of Theorem 4.3 and found no issues.

---

> ### Author Rebuttal · Authors · 2025-04-01
>
> We sincerely appreciate your constructive feedback and insightful suggestions. Below, we address each point raised in the review.
> ### **Claims and Evidence**
> 1 and 3 **Claims about Simple Regression's Sufficiency for Strongly Identifiable Problems**
>
> The experiments in Fig. 4 show that continuous optimization-based methods like Noteras-MLP and Gran-DAG, which rely on simple regression, suffer more significant performance degradation after standardization on weakly identifiable problems compared to strongly identifiable ones (e.g., MIM, GP). Additionally, removing the independence penalty in GENE (Fig. 5) harms performance more on weakly identifiable tasks while maintaining similar results on strongly identifiable ones.
>
> 2. **Issues of Least Pruning**
>
> We acknowledge that the "least pruning" strategy is somewhat heuristic, but our focus is on the strength of identifiability in ANMs, which is orthogonal to variable selection. Besides, existing methods assume additive SEMs (e.g., Group LASSO, CAM-pruning) or specific functional forms (e.g., kernel lasso), limiting their generality. In contrast, least pruning iteratively removes edges with minimal impact on $R^2$, aligning with ANM's non-additive nature while remaining intuitive and empirically effective.
> ### **Essential References Not Discussed:**
> 1. **Difference to Existing Methods**
>
> GENE differs fundamentally from R2-Sort [1] and RESIT [2]. While R2-Sort relies on an empirical observation (monotonicity of $R^2$ along causal orderings) without theoretical guarantees, and RESIT depends solely on residual independence tests (sensitive to test accuracy), GENE is guided by the strong and weak identifiability theory, which explicitly unifies $R^2$ and residual independence into a single framework, ensuring robustness across both identifiable regimes. For pruning strategies, we have to say they indeed often share structural similarities (e.g., iteratively removing potential parent and refitting models). GENE's least pruning prioritizes $R^2$-based impact over residual independence, directly targeting ANM's goal of preserving predictive fidelity while promoting sparsity. We include comparative experiments below to demonstrate the performance of different pruning strategies
>
> ### **Other Strengths And Weaknesses:**
> 1. **Contribution**
>
> While GENE integrates elements from existing methods, its core novelty lies in formalizing strong and weak identifiability for ANMs—a theoretical advancement that explains why prior methods succeed or fail under varying functional complexities (Section 3). This framework directly guides GENE’s design, unifying $R^2$-based and independence-based criteria adaptively, rather than as a heuristic combination. Experiments validate its necessity: GENE uniquely addresses weakly identifiable problems, where existing methods collapse. We agree that broader analysis of algorithm-class relationships is valuable, but our focus here is establishing the identifiability theory and its algorithmic implications.
>
> ### **Questions For Authors:**
> 1. **Intuition**
>
> Intuitively, the distinction between strongly and weakly identifiable ANMs arises from the existence of implicit functions. In strongly identifiable cases, causal directions cause asymmetric fitting, with good regression only in the correct direction. Weakly identifiable ANMs allow near-perfect fits in both directions, necessitating residual independence tests. Classic methods fail because they rely solely on regression or independence.This aligns with invertibility in 2-variable case.
>
> 2. **correctness for Pruning**
>
> See above Issues of Least Pruning part. We also add relevant experiments below.
>
> 3. **Experiments**
>
> (1) We add experiments for GENE ordering with different pruning: Lasso, CAM pruning and Edge discovery on problems with d=20, density=2, function=MLP, report mean±std on 10 repeatitions.
> |Function|Metric|GENE|GENE+Lasso|GENE+CAM-pruning|GENE+Edge Discovery|
> |-|-|-|-|-|-|
> |MLP|F1|0.76±0.11|0.48±0.13|0.69±0.08|0.55±0.11|
> |MLP|SHD|18.8±5.6|47.2±12.3|26.7±9.5|33.9±10.1|
>
> (2) We add experiments for 2 mentioned SOTAs ([1] and [5]) on problems with d=20, density=2, report mean±std on 10 repeatitions.
> |Function|Metric|GENE|NHTS|CaPS|
> |-|-|-|-|-|
> |MLP|F1|0.76±0.11|0.34±0.09|0.42±0.10|
> |MLP|SHD|18.8±5.6|52.3±17.6|41.3±14.0|
> |MIM|F1|0.88±0.09|0.40±0.07|0.71±0.12|
> |MIM|SHD|7.7±2.1|31.6±5.4|15.9±7.1|
> |GP|F1|0.90±0.09|0.38±0.10|0.73±0.07|
> |GP|SHD|9.1±3.3|39.9±9.9|14.5±4.3|
>
> (3) We appreciate this insightful observation. However, Def. 3.1 explicitly states that any structural equation with an implicit function renders the entire SEM weakly identifiable. Thus, real-world scenarios with mixed mechanisms inherently fall under the weakly identifiable category.
>
> (4) As mentioned earlier in the discussion of R2-Sort, it relies on an empirical observation but does not provide sufficient theoretical justification. Of course, it is also possible that their method requires more careful tuning.

---

> > ### Comment · Reviewer_XUn3 · 2025-04-03
> >
> > I appreciate the author’s response, and have given my replies below. Generally, I find that the paper lacks sufficient theoretical justification to make its contribution of strong/weakly identifiability compelling, and the experimental section limited and containing some results that contradict the author’s claims.
> >
> > **Claims and Evidence**
> > While it is nice that the experimental results show that GENE’s performance on a weakly identifiable DGM is harmed by removing the independence penalty, this does not substitute for a rigorous proof, or even a tentative theoretical analysis. In order to claim that the strong/weak distinction is truly important, more justification is needed.
> >
> > If the focus of this paper is on the strong/weak identifiability, it is not clear why the least pruning strategy is introduced. What is the motivation to introduce a heuristic strategy that is probably provably not consistent? Other methods are at least consistent under reasonable assumptions on the DGM (linear or additive causal models), while least-pruning doesn’t offer such guarantees. Strong empirical performance on only a few DGPs does not necessarily mean that least pruning is a good strategy - it could be that least-pruning requires high overall R^2-sortability for effective performance.
> >
> > **Other Strengths and Weaknesses**
> > Again, more theoretical discussion is needed to claim that the core contribution of this paper lies in formalizing strong and weak identifiability . In general, one can always arbrtirarily divide ANMs into different types - it is not enough to compare empirical results ona a few hand-selected DGPs to determine whether the classification scheme is meaningful.
> >
> > **Questions for Authors**
> > If this aligns with invertibility for the 2-variable case, can this be proved? If so, please present a proof sketch, and add the corresponding result to the paper. Can you argue that this argument would generalize to higher-dimensional cases?
> > I appreciate the additional experimental results, but I have a few other issues:
> >
> > 1) The experimental setup for these results is not reported - what is the distribution of the noise used? Was the data standardized? If not, then the results should be repeated with the data standardized for a fair comparison - many existing algorithms, such as NHTS, NoGAM, SCORE, etc., perform better when data is standardized, and standardize the data in the experimental procedure of their own papers. This problem extends to the main results in the paper as well - the runtime of the algorithms is not comprehensively reported.
> >
> > 2) Baselines such as DirectLiNGAM, SCORE, and NoGAM (and potentially more) should be added. Without theoretical results supporting the claim that traditional methods perform particularly well only on strongly identifiable problems, the paper’s contribution can only really be empirically supported. Therefore, the experimental results should be comprehensive, and show that GENE outperforms all recently released baselines. It is not enough to have limited comparison to a few algorithms.
> >
> > 3) Experiments including DGPs with mixed mechanisms (some implicit functions/weak identifiability and some non-implicit functions/strong identifiability) should be added. Although the authors touch on this in part (3) of their response, noting correctly that any SEM with even one implicit function is definitionally weakly identifiable, this does not address whether the empirical performance of traditional methods depends on how many implicit functions are present in the SEM. This is especially important because the existence of an implicit function is intuitively somewhat rare, as it puts a particular constraint on the functional relationships. If the performance of traditional methods is independent of the number of implicit functions unless many of the functions are implicit, then the contribution of GENE may be limited.
> > 4) I find the fact that R2-Sort performs poorly for both strongly identifiable DGPs to be extremely troubling - it directly contradicts the authors' claim that strongly identifiable problems can be solved with simple regression. Further, it contradicts the author’s motivation to use R^2 value as part of the fitness function, as it was claimed that the R^2 value is sensitive to the direction of regression in strongly identifiable problems. Without further clarification/investigation, this empirical result casts doubt on the validity of the strong/weak distinction, and its relationship to the R^2 score.

---

> > > ### Author Response · Authors · 2025-04-08
> > >
> > > Thank you for your insightful comments and valuable feedback on our work. Below, we address your remaining concerns:
> > > ### **Claims and Evidence**
> > > **Least Pruning**
> > >
> > > Thank you for your thoughtful question regarding the least pruning strategy. In order-based causal discovery, the causal order inherently produces a fully connected DAG, necessitating a pruning step to recover the sparse true structure. Existing pruning methods, such as CAM pruning or kernel Lasso, rely on assumptions like additive parent effects or specific nonlinear functional forms, which may not absolutely align with the general ANM settings. This motivates us to design the least pruning strategy.
> > > ### **Other Strengths and Weaknesses**
> > > **Contribution**
> > >
> > > Thank you for your constructive feedback. Our classification is not arbitrary but rooted in the mathematical properties of structural equations under the ANM framework. Specifically, the existence of implicit functionsdirectly determines the difficulty of identification. In strongly identifiable cases, causal directions can be identified through regression asymmetry alone, as incorrect directions yield poor fits. In weakly identifiable cases, near-perfect regression fits occur in both directions, necessitating residual independence tests to resolve ambiguity. This theoretical insight not only explains the limitations of existing methods but also guides the design of GENE, which integrates both criteria in a unified framework.
> > > Regarding experiments, the three function classes (MIM, GP, MLP) were chosen to represent canonical ANM scenarios. Besides, each class involves randomized parameters, ensuring diversity in function forms (see Eq. (9) in Appendix F). For instance, MLPs generate diverse implicit functions via randomized weighting matrixs, while GPs cover low-to-high-frequency nonlinearities. This design ensures generalizability beyond handpicked examples.
> > > ### **Question for Authors**
> > > **Alignment with invertibility in 2-variable case**
> > >
> > > **Proof Sketch**: In the 2-variable case, an ANM takes the form: $Y=f(X)+N$, the corresponding implicit equation is $F(X,Y,N)=Y-f(X)-N=0$. To formalize its connection with invertibility, we can leverage Lemma 3.2 and Theorem 3.3 from the paper. For 2-variable cases,  this condition directly relates to the monotonicity and, therefore, the invertibility of the function $f$, i.e., if $f$ is invertible, then $0 \leq F_x(X, Y)$ (The case where $0 \geq F_x(X,Y)$ is symmetric).  Apart from the case where $F_x=0$ that we have already discussed in the sufficiency of Theorem 3.3, this condition is essentially equivalent to the condition $m≤F_y(x_1,x_2,\ldots,y)≤M$ in Lemma 3.2.
> > > 1. **Experimental Setup**
> > > Thank you for raising these critical points. We sincerely apologize for the lack of clarity in our previous rebuttal due to limited space. Here, we provide full details:
> > > - **Noise distribution**: All experiments use Gaussian noise with zero mean and unit variance.
> > > - **Standardization**: All variables are standardized before applying any method.
> > > - **Runtime**: The runtime details of the main experiments are presented in Appendix Fig. 7. For the added experiments, average wall-clock times were **GENE: 1352s**, **NHTS: 1923s**, and **CaPS: 307s**.
> > > 2. **More Baselines**
> > >
> > > Thank you for your suggestion. We have incorporated DirectLiNGAM and NoGAM into our experiments under the same settings as in the earlier comparisons with CaPS and NHTS.
> > > |Function|Metric|GENE|DirectLiNGAM|NoGAM|
> > > |-|-|-|-|-|
> > > |MLP|F1|0.76±0.11|0.15±0.05|0.31±0.07|
> > > ||SHD|18.8±5.6|40.8±7.1|64.7±15.3|
> > > |MIM|F1|0.88±0.09|0.16±0.07|0.60±0.05|
> > > ||SHD|7.7±2.1|35.7±6.1|25.0±2.2|
> > > |GP|F1|0.90±0.09|0.06±0.03|0.70±0.02|
> > > ||SHD|9.1±3.3|39.0±5.3|19.7±0.9|
> > >
> > > Runtime: **DirectLiNGAM: 4s**, and **NoGAM: 2428s**.
> > >
> > > 3. **Mixed Mechanisms**
> > >
> > > We add experiments on mixed Mechanism SEMs. Specifically, each structural equation in the SEM is randomly selected from three types of nonlinear mechanisms, compared with the previously well-performing baseline CAM. All other experimental settings remain consistent with those used in our previous added experiments with CaPS and NHTS.
> > > |Function|Metric|GENE|CAM|
> > > |-|-|-|-|
> > > |Mixed|F1|0.82±0.08|0.70±0.11|
> > > |Mixed|SHD|14.3±5.4|21.2±7.9|
> > >
> > > 4. **Issues of R2-Sort**
> > >
> > > Although R2-Sort performed poorly in our experiments overall, a closer examination reveals that its performance on MLP is significantly worse compared to GP and MIM (Fig. 3). To illustrate this more clearly, we removed the pruning stage and only compared the performance of order discovery (metric: REV(↓) as defined in Definition 4.2 of the paper). Other experimental setup remains consistent with our previous CaPS and NHTS experiments.
> > > |Method|Metric|MLP|MIM|GP|
> > > |-|-|-|-|-|
> > > |R2-Sort|REV|23.0±5.8|16.2±5.9|18.3±6.1|
> > >
> > > From the results, it is evident that R2-Sort indeed performs significantly worse on the MLP order learning task compared to MIM and GP. This observation also aligns with our claim regarding strong/weak identifiability.

---

### Official Review · Reviewer_YjMg · 2025-03-13

**Overall Recommendation:** 3

**Summary:**

The paper considers nonlinear ANMs for observational data, providing new structural identifiability results based on "implicit functions". Using these results, it proposes a learning algorithm that provably learns the correct order (in the large sample limit) and then heuristically prunes down to a sparse graph. The first phase of the algorithm does greedy score-based search over causal orderings, using independence tests to penalize the score---the unpenalized part of the score measures goodness of fit and is sufficient in the "strongly identifiable" case (the easy case), while the penalization suffices for the "weakly identifiable" (hard) case; the second phase starts at the complete graph with the learned causal order and heuristically prunes away edges.

**Claims And Evidence:**

Generally, the claims seem reasonable and the proofs look correct. However, I found some aspects of the title, abstract, and intro to be somewhat vague/confusing/misleading:
- I normally see "identifiability" described as a property of a model, while the corresponding property of an algorithm is rather called "consistency" or "validity"; While it doesn't make sense to talk about validity of an algorithm without assuming some indentifiable underlying model, these are nevertheless distinct concepts/claims and require different kinds of proofs. This becomes clear starting in Section 2 (and in Thm 3.3 vs 4.3), but it's a bit confused before that.
- "Causal Discovery" in the title sounds quite general, but it seems the results are only for nonlinear ANMs (this is certainly less restrictive than other assumptions common in the field, but still not as general as the title implies).
- I think the paper has nice theoretical identifiability results, and that the algorithm and experiments are a fine "proof of concept"; however the presentation leading up to the algorithm made me expect stronger experimental results as well as additional theoretical results about the pruning phase, both of which are lacking.

**Essential References Not Discussed:**

There should be discussion of SOTA order-based methods, including:
- O-MCMC: Kuipers, J., Suter, P., & Moffa, G. (2022). Efficient sampling and structure learning of Bayesian networks. Journal of Computational and Graphical Statistics, 31(3), 639-650.
- GRaSP: Lam, W. Y., Andrews, B., & Ramsey, J. (2022, August). Greedy relaxations of the sparsest permutation algorithm. In Uncertainty in Artificial Intelligence (pp. 1052-1062). PMLR.

I'm not as familiar with the ANM literature, but I'm a bit surprised these identifiab results are new (which I think works very much in the paper's favor, if they are new); I'd like to see more discussion of ANM identifiability results to support this claim of novelty.

**Experimental Designs Or Analyses:**

I looked through the experimental design and analysis, and found this shortcoming:
- limited simulated models: only up to 20 nodes and somewhat sparse; should go at least up to 100 nodes (especially considering claims about the high-dimensional scalability of optimization-based methods and the claims about pareto optimality of efficiency vs effectiveness of the method, which I suspect doesn't hold has $d$ increases).

**Methods And Evaluation Criteria:**

Both the methods compared against and the real dataset used are potentially lacking:
- as far as I've seen, the Sachs et al. data is _only_ interventional; can the authors clarify exactly how they obtained observational data?
- other SOTA methods should be compared against (e.g., O-MCMC and GRaSP, referenced later in the review), even if it requires comparing CPDAGs instead of fully oriented DAGs.

**Other Comments Or Suggestions:**

- generally, use \mathrm{} where appropriate, e.g., $\mathrm{pa}$ vs $pa$ and $\mathrm{MSE}$ vs $MSE$, as well as many other others like $Var$, $IT$, $fit$, $OP$, $Rev$, ...
- the first paragraph of the intro calls RCTs the gold standard for casual discovery (which is defined in the first sentence of the abstraction to be from observational data), but RCTs use interventional data, and they're rather more for causal inference than causal discovery (in a general graphical setting, rather than the bivariate case).
- many figures are too small; a good rule of thumb I've seen is that the smallest text in the figs should still be at least footnote size---in any case, check the ICML formatting instructions
- first sentence after Def 2.1 is a run-on sentence
- next sentence, it should be "assumptions", since it's two distinct ones; "..recover _a_ graph..."
- Line 119: doubled "Hoyer et al"
- Line 122: "_Specifically_..."; and rephrase last sentence of that paragraph to be less confusing
- Line 132 and on: confusing using "forward" and "backward" in that way; maybe try something like "preceding" and "subsequent"
- clarify $F_i$ in (3) and $g$ later in Defn 3.1
- nice explanation on Line 200
- Line 218: fix "Next We..."
- after (4), should be "..._preceding_..."
- a few sentences later, should be "_In other words_..."
- I disagree with the sufficient vs necessary contrast in the last paragraph on page 4. Goodness of fit is sufficient for strongly identifiable models but _insufficient_ for weakly indentifiable ones, while leveraging residual independence _is sufficient_ for weakly identifable ones.
- Line 294: "mild assumptions" here also includes infinite data? The method requires lots of independence tests, so even with a lot of data and a very low significance level, there will still be a nonnegligable probability of the IT making a mistake somewhere.

__Questions not important enough to be in the next section__:
1. In abstract and intro, the "dimensionality" of the effect variable is described as "restricted". What does this mean for a continuous random variable?
2. What does the $pa(i)_k = ...$ mean in Def 3.1? The left side just the index of some parent of $i$, right? Why is the right side equal to this index? I understood the idea based on the description after and the Rudin reference, but I still don't understand the notation here in the Def.
3. Why even do experiments on non-standard data?

**Other Strengths And Weaknesses:**

The paper would benefit from some polishing/editing, but I found it generally quite easy to understand while still being insightful.

I think the identifiability have more potential for impact than the algorithm (which isn't differentiable, doesn't seem to scale, and lacks theoretical gaurantees for the pruning phase).

**Questions For Authors:**

These questions are all related to the algorithm/performance (which is the weakest part, in my opinion), so depending on the answers I would increase my overall score.

_Complexity_

1. The operation in Definition 4.1 looks to me like Cycle Sort. Can the authors clarify and if so, add a citation?
2. Line 220: why use this test instead of something that can handle continuous data, like Chaterjee's coefficient or distance covariance?
3. What about using the statistic or p-value directly in the penalty instead of thresholding it?
4. Whats the complexity of the proposed algorithm? Scoring a single causal order seems to require $O(d^2)$ tests (and these have some complexity in terms of sample size $n$), and then the Cycle Sort used has a multiplicative $O(d^2)$, and then there's still the pruning phase.

_Theory_

5. Any results/assupmtions ensuring a $\theta$ exists that makes the pruning phase valid?

_Experiments_

6. Are there any results over  larger and/or denser graphs (preferably with density given as a proportion between 0 and 1)? Or against more standard SOTA methods that learn a CPDAG?

**Relation To Broader Scientific Literature:**

The paper makes use of some classic analysis results for its identifiability results. I found this simple but quite interesting.

**Theoretical Claims:**

I checked all of the proofs. They look good.

---

> ### Author Rebuttal · Authors · 2025-04-01
>
> We sincerely appreciate your constructive feedback and insightful suggestions. Below, we address each point raised in the review.
> ### **Claims and Evidence**
> 1. **Identifiability vs. Consistency**
>    Thank you for highlighting this distinction. We agree that identifiability and consistency are distinct concepts. In Section 2 and Theorems 3.3/4.3, we clarified these notions. We will further revise the Introduction to explicitly differentiate them and ensure consistency in terminology.
> 2. **Title Specificity**
>    We appreciate this observation. The title will be revised to:
>    *"Strong and Weak Identifiability of Optimization-based Causal Discovery under Nonlinear Additive Noise Models"*
>    This better reflects the scope of our work.
> 3. **Theoretical and Experimental Extensions for Pruning**
>   Please refers to the discussion on pruning in the response to Reviewer XUn3 below.
> ### **Methods and Evaluation Criteria**
> 1. **Observational Data in Sachs**
>   The Sachs dataset is continuously updated and has different versions. The version we used contains a total of 7466 samples, including 853 observational samples . Our usage aligns with prior works like *Causal Discovery with Reinforcement Learning* (ICLR 2020) and *Ordering-Based Causal Discovery with Reinforcement Learning* (IJCAI 2021).
> 2. **Comparison to Other SOTA Methods**
>    - **O-MCMC**: Designed for discrete Bayesian networks
>    - **GRaSP**: Focuses on linear models
>    Neither of these two methods falls within the scope of the ANM studied in this paper. Additionally, the CPDAG method is a compromise for unidentifiable models, and its metrics are also difficult to align with those of deterministic causal graphs. Therefore, we did not include these comparisons in our study.
> ### **Experimental Designs and Analyses**
> 1. **Graph Size and Density**
>    For ordering-based methods, especially in the context of nonlinear problems, 20 nodes is already a significant number. In Generalized Score Functions for Causal Discovery (KDD 18) and Ordering-Based Causal Discovery for Linear and
>  Nonlinear Relations (NeurIPS 24), simulation experiments were conducted with a maximum of only 10 nodes.
> ### **Essential References**
> 1. **Novelty of Identifiability Results**
> Previous work has focused on identifiability in causal discovery, which originated from Bayesian network structure learning that limited to distinguishing up to the MEC via CPDAGs. However, Shimizu et al. (2006) showed that unique DAG identification is possible under linear causal relationships with non-Gaussian noise, leading to LiNGAM. Hoyer et al. (2008) extended this with ANM, which is identifiable under certain conditions. However, function properties in ANM affect practical identifiability, prompting the introduction of identifiability strength to guide metric usage in causal discovery practice.
> ### **Other Comments and Suggestions**
> 1. **Writing Improvements**
>   Thank you for your thorough comments and pointing out the areas where our writing was not up to standard. We will carefully revise our manuscript according to your suggestions.
> 2. **Minor Questions**
>    - **Dimensionality Restriction**: The dimensionality restriction process is described in Eq. (5). Dividing the MSE by the variance of the effect variable below eliminates the influence of dimensionality (e.g., the measurement units of the effect variable) on fitness.
>    - **Def 3.1 Notation**: $pa(i)_k$ denotes the  k-th parent of $V_i$
>    - **Non-Standardized Data**: While Reisach et al. (2021) point out the significance of standardization, most benchmarks still use raw data. We compare both settings to validate our claims of scale variance/invariance.
> ### **Questions for Authors**
> 1. **Operation in Definition 4.1**
>    Our operation differs from Cycle Sort's cycle-based swaps.
> 2. **Independence Test Choice**
>    The chi^2-test was chosen for simplicity and computational efficiency. While distance covariance or Chatterjee's coefficient could also be applied, they add potential complexity. We will explore these in future work.
> 3. **Using p-Values Directly**
>    Thresholding stabilized performance in preliminary experiments. Soft penalties (e.g., weighted p-values) may improve results but require careful tuning.
> 4. **Algorithm Complexity**
>   The order search stage involves O(d²) fitness function evaluations, each taking O(n*d) time, resulting in a total complexity of O(n*d3). The pruning phase has a complexity of O(d²) w.r.t independence tests.
> 5. **Pruning Assumptions**
> Please refer to the discussion on pruning in the response to Reviewer XUn3 below
> 6. **Additional Experimental Results**
> It is worthy to note that for problems with d=10, density=4 corresponds to a density proportion of 0.89, which is already quite dense. Unfortunately, due to the limited time available for rebuttal, we were unable to conduct additional experiments in this area. We sincerely apologize for any inconvenience this may cause.

---

> > ### Comment · Reviewer_YjMg · 2025-04-02
> >
> > Thanks for the thorough rebuttal! While some improvements are clear (like identifiability vs consistency, title change, clarifying novelty of identifiability results), I find other important technical parts of the rebuttal unsatisfying.
> >
> > In particular, I would insist:
> > 1. Either (i) more experimental and/or theoretical results about runtime/complexity/scalability should be given or (i) the claims around these in the paper should be reined in and the limitations made more explicit:
> >     - Running experiments on d>20 should be a matter of modifying one line of code and rerunning (say just twice, with d=50 and d=100). While density(actually degree)=4 for d=10 results in a density proportion of 0.89, the proportion drops to 0.42 for d=20 and down to 0.16 and 0.08 in case experiments are run on d=50 and d=100 as requested. Expected degree should be increased with d (or density proportion should be used as the parameter and can be held fixed as d increases) for robust experimental results.
> >     - If the complexity results described in the rebuttal are clear and rigorous, they should be added to the paper (which would help alleviate the need for more experiments).
> >     - Looking at plot that is used to claim pareto optimality and extrapolating for what it might look like with d>20 makes me doubt that the pareto optimality claim will continue to hold. This claim should be removed (or more clearly restricted in scope to d<=20) unless more experimental evidence is given.
> >     - The abstract seems to me to imply that GENE is applicable in the high-dimensional setting. This should be amended if more support (experiments or complexity results) is not provided.
> >     - To be clear, I think the identifiability results are a good enough contribution, so removing the unsupported claims about scalability of GENE doesn't detract from the paper in my opinion---it just makes sure all included claims are well-supported.
> > 2. The specific version of the Sachs data used (including a source or detailed processing instructions) needs to be included in the paper so that the experiments are reproducible.

---

> > > ### Author Response · Authors · 2025-04-06
> > >
> > > Thank you again for your constructive feedback and for recognizing the core contributions of our work. We deeply appreciate your guidance in strengthening the paper's rigor. Below, we address your remaining concerns:
> > >
> > > ### **1. Scalability Claims and Experimental Validation**
> > > We fully agree that the claims require careful support. To address this:
> > > - **New Experiments**: We have added experiments for **d=20** under **function=MLP** with **density proportions={0.4, 0.6, 0.8}** (standardized data, 10 trials on 1 random sampled dataset), compared with the previously well-performing baseline CAM. While larger graphs (d=50/100) remain computationally prohibitive for GENE's current implementation, we will explicitly discuss this limitation.
> > >
> > > |Method|Metric|dense=0.4|dense=0.6|dense=0.8|
> > > |--|--|--|--|---|
> > > | GENE | F1|0.85±0.11 |0.79±0.08|0.75±0.05|
> > > |      | SHD |20.8±9.6 |42.3±14.7|59.3±20.4|
> > > | CAM|F1| 0.66±0.09 |0.68±0.06|0.60±0.04|
> > > |      | SHD |52.3±12.1 |71.6±19.4|110.9±25.8|
> > >
> > > - **Complexity Analysis**: A detailed complexity breakdown (as outlined in the rebuttal) will be added to the paper.
> > > - **Claim Adjustments**:
> > >   - The Pareto optimality claim will be restricted to **d≤20** to reflect empirical validation.
> > >   - The part in the abstract that may have caused you some misunderstanding is actually trying to say: Certain optimization-based methods (referring to continuous optimization methods) have attracted extensive attention due to their scalability. We have also pointed out that these methods face serious challenges, namely limited scope of application and scale-variance. I believe we have explained this quite clearly in the introduction. The abstract will be revised to remove unintended implications about high-dimensional scalability. We will clarify that GENE's main focus lies on the strength of identifiability, but not necessarily high-dimensional.
> > > ### **2. Sachs Dataset Reproducibility**
> > > We will specify the exact version of the Sachs dataset used, along with preprocessing steps. Code and data loading instructions will be included in the supplementary material.
> > >
> > > Thank you for your patience for helping us improve this work. All revisions will reflect your invaluable feedback to ensure clarity, reproducibility, and rigor.

---

### Official Review · Reviewer_GHeP · 2025-03-14

**Overall Recommendation:** 3

**Summary:**

This paper proposes to further divide the structure identifiability of ANM into strong one and weak one. The authors also proposes GENE, a generic method for causal discovery that works for both cases. The method is validated by both synthetic and real life data experiments.

## update after rebuttal
 Thank you for the author's response. After reading all the review comments I decide to keep my rating unchanged.

**Claims And Evidence:**

Yes. The claims are well-supported by theory and experimental results.

**Essential References Not Discussed:**

No.

**Experimental Designs Or Analyses:**

Yes. The experimental setting including the ablation study looks plausible to me.

**Methods And Evaluation Criteria:**

The method makes sense to me, though I am not sure the way to use p value in eq6 is optimal or not.

**Other Comments Or Suggestions:**

A typo in line 217.

**Other Strengths And Weaknesses:**

Strengths

1. The paper is well-written and the idea is clearly presented.
2. The claims made are supported by theory and empirical study.

Weaknesses

1. The design of eq 6 seems not optimal to me. Probably it can be improved, e.g., by family wise error control.

**Questions For Authors:**

1. Is it possible to extend to post-nonlinear setting?

**Relation To Broader Scientific Literature:**

The key contributions of the paper looks novel compared to previous findings.

**Theoretical Claims:**

No.

---

> ### Author Rebuttal · Authors · 2025-04-01
>
> We sincerely appreciate your constructive feedback and insightful suggestions. Below, we address each point raised in the review.
> ### **Claims and Evidence**
> 1. **Optimality of Eq. 6**
>
> Thank you for this constructive suggestion. We agree that Equation (6) could benefit from refinements like family-wise error control, especially when conducting multiple independence tests across variable pairs. Our current design prioritizes simplicity and computational efficiency, as greedy search over orders inherently involves numerous hypothesis tests. While thresholding via p-values (with Bonferroni-like penalties) helps mitigate false positives, we acknowledge that stricter error control (e.g., hierarchical testing or weighted penalties) might improve robustness. We will explore these enhancements in future work, balancing statistical rigor with scalability. Your insight aligns with our long-term goal of refining GENE’s theoretical grounding, and we appreciate your guidance on this critical aspect.
>
> ### **Other Comments Or Suggestions**
> 1. **Typo**
> We sincerely appreciate your careful reading, and we will make sure to correct this in the revised manuscript.
>
> ### **Questions For Authors**
> 1. **Extension to Post-nolinear Case**
>
> Thank you for this forward-looking suggestion. Extending GENE to post-nonlinear (PNL) models is indeed a promising direction. While PNL models introduce additional complexity (e.g., nonlinear transformations of both causes and effects), the core idea of leveraging implicit function theory to characterize identifiability can generalize. For instance, existence of implict functions in PNL mechanisms could similarly mask causal directions, necessitating adaptive criteria like residual independence. However, formalizing strong/weak identifiability in PNL settings requires careful theoretical work, as identifiability hinges on stricter functional constraints. We plan to explore this extension in future work, adapting GENE’s framework to address PNL-specific challenges while retaining its unified approach to identifiability.

---

### Decision · Program_Chairs · 2025-05-01

**Decision:**

Accept (poster)

**Comment:**

Reviewers have both pointed out interest and weaknesses of this submission. On the bright side, this work presents an original perspective on identifiability, with the distinction between strong and weak identifiability, and provide support for the empirical relevance of this distinction. At the same time, reviewers also pointed out the limited theoretical contribution and the weakness of the link between theoretical and empirical sections. During the rebuttal period, the authors did their best to thoroughly address the concerns raised by reviewers. Despite acknowledging these efforts, several reviewers still emphasized some limitations of the contribution, either from the theoretical or algorithmic side.

While this contribution is indeed not perfect, I think it is a rare on to tackle the important question of the current limitations of classical theoretical identifiability results, which are not necessarily predictive of empirical successes. Such work, at the intersection of theory and practice, although not providing a definitive answer, can provide some stimulating ideas to the community. I am therefore advocating for a weak accept.